# Brain tropism acquisition: The spatial dynamics and evolution of a measles virus collective infectious unit that drove lethal subacute sclerosing panencephalitis

Iris Yousaf[1,2☯], William W. Hannon[3,4☯], Ryan C. Donohue[2], Christian K. Pfaller[1,2], Kalpana Yadav[1], Ryan J. Dikdan[5], Sanjay Tyagi[5], Declan C. Schroeder[6], Wun-Ju Shieh[7¤], Paul A. Rota[8], Alison F. Feder[9,10]*, Roberto Cattaneo[1,2]*

1 Department of Molecular Medicine, Mayo Clinic, Rochester, Minnesota, United States of America, 2 Mayo Clinic Graduate School of Biomedical Sciences, Rochester, Minnesota, United States of America, 3 Basic Sciences and Computational Biology, Fred Hutchinson Cancer Center, Seattle, Washington, United States of America, 4 Molecular and Cellular Biology Graduate Program, University of Washington, Seattle, Washington, United States of America, 5 Public Health Research Institute, Rutgers University, Newark, New Jersey, United States of America, 6 Department of Veterinary Population Medicine, University of Minnesota, St Paul, Minnesota, United States of America, 7 Infectious Diseases Pathology Branch, Division of High Consequence Pathogens and Pathology, Center for Disease Control and Prevention, Atlanta, Georgia, United States of America, 8 Division of Viral Diseases, National Center for Immunization and Respiratory Diseases, Center for Disease Control and Prevention, Atlanta, Georgia, United States of America, 9 Genome Sciences, University of Washington, Seattle, Washington, United States of America, 10 Public Health Sciences and Computational Biology, Fred Hutchinson Cancer Center, Seattle, Washington, United States of America

☯ These authors contributed equally to this work.
¤ Current address: Department of Microbiology and Immunology, Taipei Medical University, Taipei, Taiwan
* affeder@uw.edu (AFF); Cattaneo.Roberto@mayo.edu (RC)

**Data Availability Statement:** The raw Illumina sequencing reads are available on the NCBI

## Abstract

It is increasingly appreciated that pathogens can spread as infectious units constituted by multiple, genetically diverse genomes, also called collective infectious units or genome collectives. However, genetic characterization of the spatial dynamics of collective infectious units in animal hosts is demanding, and it is rarely feasible in humans. Measles virus (MeV), whose spread in lymphatic tissues and airway epithelia relies on collective infectious units, can, in rare cases, cause subacute sclerosing panencephalitis (SSPE), a lethal human brain disease. In different SSPE cases, MeV acquisition of brain tropism has been attributed to mutations affecting either the fusion or the matrix protein, or both, but the overarching mechanism driving brain adaptation is not understood. Here we analyzed MeV RNA from several spatially distinct brain regions of an individual who succumbed to SSPE. Surprisingly, we identified two major MeV genome subpopulations present at variable frequencies in all 15 brain specimens examined. Both genome types accumulated mutations like those shown to favor receptor-independent cell-cell spread in other SSPE cases. Most infected cells carried both genome types, suggesting the possibility of genetic complementation. We cannot definitively chart the history of the spread of this virus in the brain, but several observations suggest that mutant genomes generated in the frontal cortex moved outwards as a collective and diversified. During diversification, mutations affecting the cytoplasmic tails of both

Sequence Read Archive under the BioProject accession number PRJNA1024527. The patient specific reference sequence is available on GitHub at https://github.com/jbloomlab/MeV_SSPE_Dynamics/blob/main/config/ref/MeVChiTok-SSPE.fa. The code used to perform all analysis in the paper is available on GitHub at https://github.com/jbloomlab/MeV_SSPE_Dynamics. The repository is also archived on Zenodo at DOI 10.5281/zenodo.8412085.

**Funding:** This project was supported by grants AI159230 and AI143791 to RC, and CA227291 to ST. IY was supported in part by the Mayo Clinic Graduate School of Biomedical Sciences, and RJD by the New Jersey Alliance for Clinical and Translational Science TL1. The funders had no role in study design, data collection and analysis, decision to publish, or preparation of the manuscript.

**Competing interests:** The authors have declared that no competing interests exist.

viral envelope proteins emerged and fluctuated in frequency across genetic backgrounds, suggesting convergent and potentially frequency-dependent evolution for modulation of fusogenicity. We propose that a collective infectious unit drove MeV pathogenesis in this brain. Re-examination of published data suggests that similar processes may have occurred in other SSPE cases. Our studies provide a primer for analyses of the evolution of collective infectious units of other pathogens that cause lethal disease in humans.

## Author summary

Autopsy material from the brain of a patient who succumbed to subacute sclerosing pan-encephalitis (SSPE), a lethal brain infection caused by measles virus (MeV) persistence, provided a unique opportunity to characterize the spatial dynamics of a collective infectious unit in a human host. We discovered that brain colonization was driven by multiple distinct genome lineages that co-replicated even at the level of single cells. Brain adaptation yielded a genetically diverse and widely dispersed viral genome population at patient death. We identified mutations affecting the matrix and fusion proteins similar or identical to those previously shown to drive brain spread in other SSPE cases. Mutations affecting the cytoplasmic tails of both envelope proteins–fusion and hemagglutinin–appeared to be constrained to intermediate prevalence by frequency-dependent selection, which may permit the virus to achieve optimal fusogenicity for brain spread. These observations are best interpreted by postulating the spread of an evolving collective infectious unit constituted by multiple genetically diverse genomes. Our results raise profound questions about the importance of collective infectious units in human disease.

## Introduction

Acute viral infections are typically cleared by the host's innate and adaptive immune responses, but even non-integrating RNA viruses can persist [1,2]. Neurons of the central nervous system are a privileged location for persistence because the host cannot deploy the cytolytic and inflammatory defense mechanisms that control infections in renewable cell types [3,4]. Sub-acute sclerosing panencephalitis (SSPE) provides the prime example of a persistent brain infection caused by a human RNA virus. SSPE, which occurs in about 1 in 10,000 individuals typically 5–10 years after they experience an acute infection as a child [5–7], starts with subtle signs of intellectual and psychological dysfunction and progresses to sensory and motor function deterioration that ultimately leads to death [8,9]. There are no effective treatments for SSPE, however nonspecific antivirals (interferons, ribavirin, and inosine pranobex) have been used [10]. Although vaccination against measles prevents SSPE, this lethal disease is resurging due to vaccine hesitancy and missed immunizations due to COVID-19 related disruptions [11,12].

In the brain, MeV genomes spread, presumably trans-synaptically, without assembling infectious particles or forming visible syncytia [13–15]. This occurs even though neither of the canonical MeV receptors, signaling lymphocytic activation molecule (*SLAMF1*) or nectin-4, are expressed [16,17]. In the absence of these receptors, the membrane fusion apparatus is activated by brain-specific isoforms of the cell adhesion molecules CADM1 and CADM2 when it reaches the plasma membrane of infected cells [18,19].

The ability of MeV to spread in this manner is the result of mutations affecting the fusion (F) and the matrix (M) genes. In more than half of SSPE cases, mutations impair the M protein particle assembly organization function [20–31]. In addition, in almost every SSPE case mutations alter the F protein function: certain mutations destabilize the ectodomain and allow receptor-independent fusion activation [32–35], while others truncate the cytoplasmic tail, disconnecting F from fusion inhibition exerted by the M protein [36–38]. Brain injections of recombinant MeV in rodent models have confirmed the relevance of these specific classes of mutations in neuropathogenesis [39,40]. Both MeV lacking a functional M protein and MeV with a truncated F protein cytoplasmic tail lost acute pathogenicity but penetrated more deeply into the brain parenchyma than standard MeV [41,42]. However, animal models do not faithfully replicate the selective environment of the human brain [43].

Alternatively, the events driving MeV spread in the human brain could be reconstructed through high coverage sequencing data of complete MeV genomes collected from different regions of the brain. However, when SSPE cases were more prevalent, sequencing technology was in its early stages, and only partial sequences of some genes were obtained from a limited number of cases [24–34]. The widespread adoption of the measles vaccine almost eliminated SSPE, diminishing the likelihood of obtaining autopsy material capable of providing complete coverage of MeV genomes replicating in multiple brain regions. Thankfully, a frozen SSPE brain autopsy was donated to the Center for Disease Control and Prevention, making this analysis possible.

We analyzed MeV RNA from 15 spatially distinct brain regions of an individual who succumbed to SSPE, both by deep sequencing and at the single cell level. The combined sequencing data from all brain specimens covers the 15,894 bases MeV genome 0.89 million times. We made the following insights into SSPE progression in this brain. First, viral replication was extensive in most regions. Second, multiple lines of evidence support the initiation of brain spread in the frontal cortex. Third, in all 15 brain specimens analyzed, we detected not just one, but two distinct major MeV genome subpopulations, each showing extensive spatially restricted diversification. Lastly, during brain adaptation, putative driver mutations affecting the cytoplasmic tails of both envelope proteins–F, and hemagglutinin (H)–fluctuated in frequency across regions, suggesting convergent evolution for modulation of fusogenicity.

## Results

### Robust MeV transcription in two brain specimens

A US resident born in Central America succumbed to SSPE when he was 24-years old. At autopsy, the entire brain was frozen and donated to the Center for Disease Control and Prevention (CDC). Two specimens (SSPE1 and SSPE2) were removed from the surface of the frozen brain for a pilot analysis. RNA quality was adequate, as demonstrated by partial preservation of the 28S and 18S ribosomal bands (**Fig 1A and 1B,** left panels).

The presence of MeV transcripts and genomic RNA was confirmed using specific probes (**Fig 1A and 1B**, right panels). Probe N(+) detects the 2 kilobases (kb) nucleocapsid (N) mRNA, and the 3.5 kb N-P mRNA which also includes the phosphoprotein (P) gene. The complementary strand probe L(-) detects 16 kb negative strand genomes and shorter defective genomes. N(+) analyses of both SSPE1 and SSPE2 documented robust N transcription, reaching similar levels as in a control infection of HeLa cells. L(-) analyses detected full-length genomes in both SSPE1 and SSPE2, and shorter molecules that may represent defective genomes.

We assessed the relative amount of viral and cellular RNA in both SSPE specimens by RNA sequencing after depletion of ribosomal RNA. Roughly 15% of the non-ribosomal RNA in

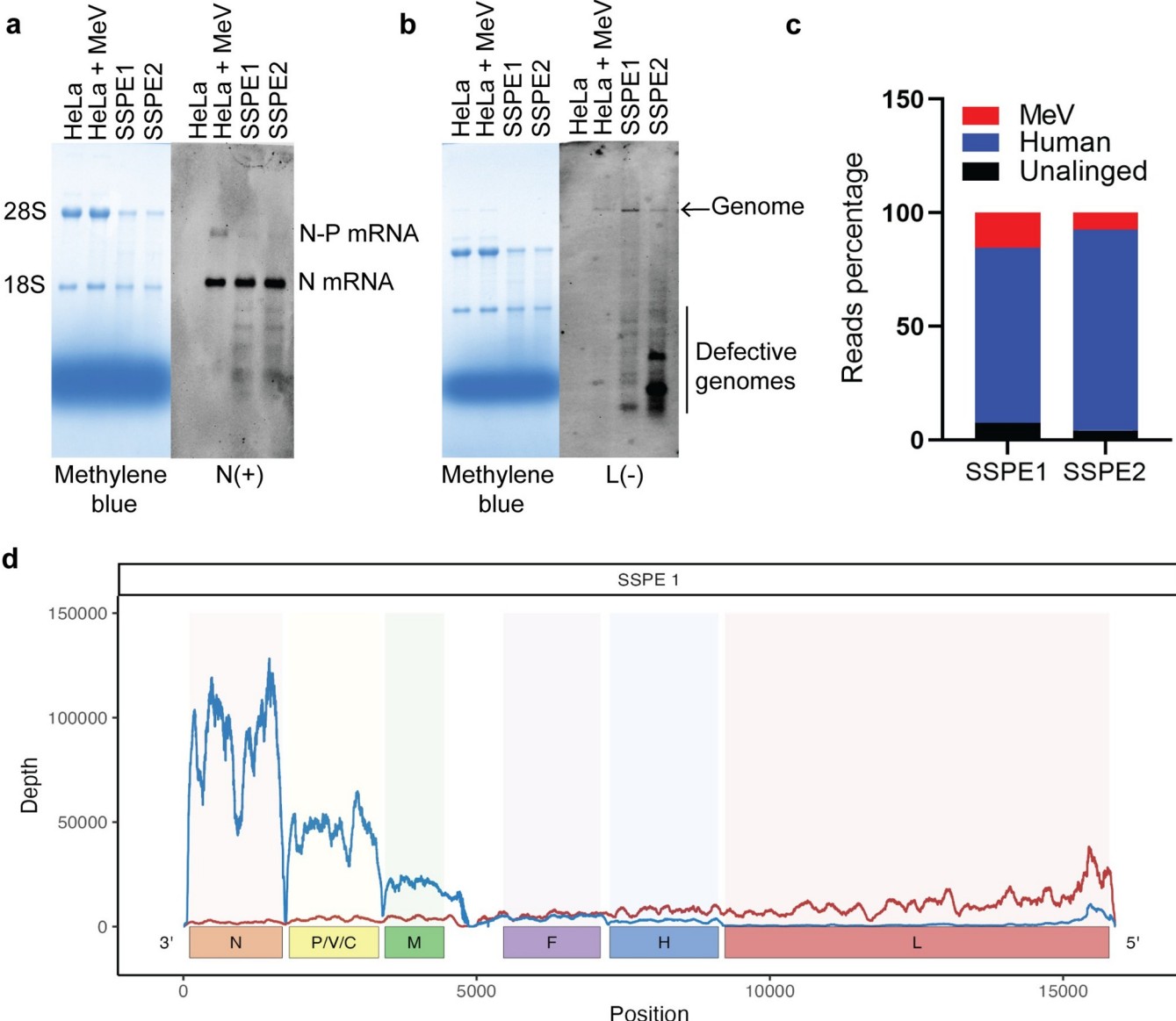

**Fig 1. Robust MeV replication and transcription in two brain specimens.** (A, B, left panels) Methylene blue stained RNA gels comparing the integrity of RNA extracted from SSPE brain specimens to that of HeLa cells uninfected or infected with a MeV vaccine strain. (A, B, right panels) Northern blots of the gels probed using (A) a probe detecting positive strand MeV N (monocistronic) and N-P (dicistronic) mRNAs or (B) a probe detecting negative sense genomic RNA. (C) Pie chart showing the number of reads that aligned to MeV genome, human genome (release #38) and unaligned reads in specimen SSPE1 and SSPE2. (D) MeV genome coverage plot showing the positive (blue line) and negative (red line) strand reads in specimen SSPE1. x-axis shows schematic of MeV genome in negative sense orientation and y-axis represents reads per nucleotide.

SSPE1, and 8% of the non-ribosomal RNA in SSPE2, was of viral origin (**Fig 1C**). For comparison, the peak level of MeV RNA in HeLa cells is about 25% [24].

Analyses of the polarity and distribution of the sequencing reads showed MeV transcript levels decreasing in concert with the distance of the six genes from the 3' end of the negative strand genome (**Fig 1D**, blue line), reflecting transcriptional attenuation at gene junctions, as observed in lytic infections [24]. The negative strand reads were more evenly distributed except for an accumulation near the 5' end of the genome (**Fig 1D**, red line, peaks at right) as observed in some lytic infections and consistent with the presence of short defective genomes [44].

### Two distinct genome populations in both specimens

We then analyzed the MeV genomes replicating in specimens SSPE1 and SSPE2. Ideally, mutations are identified by comparison with the sequence of the virus that infected the individual. However, this information is not available. To overcome this limitation, since diagnostic sequencing identified a D3 genotype, we generated a reference genome sequence including information from D3 genomes circulating at the time of infection.

We used this reference to identify single nucleotide variants (SNVs) in each sample down to 2% frequency. **Fig 2** illustrates the frequency and genomic location of the nucleotides

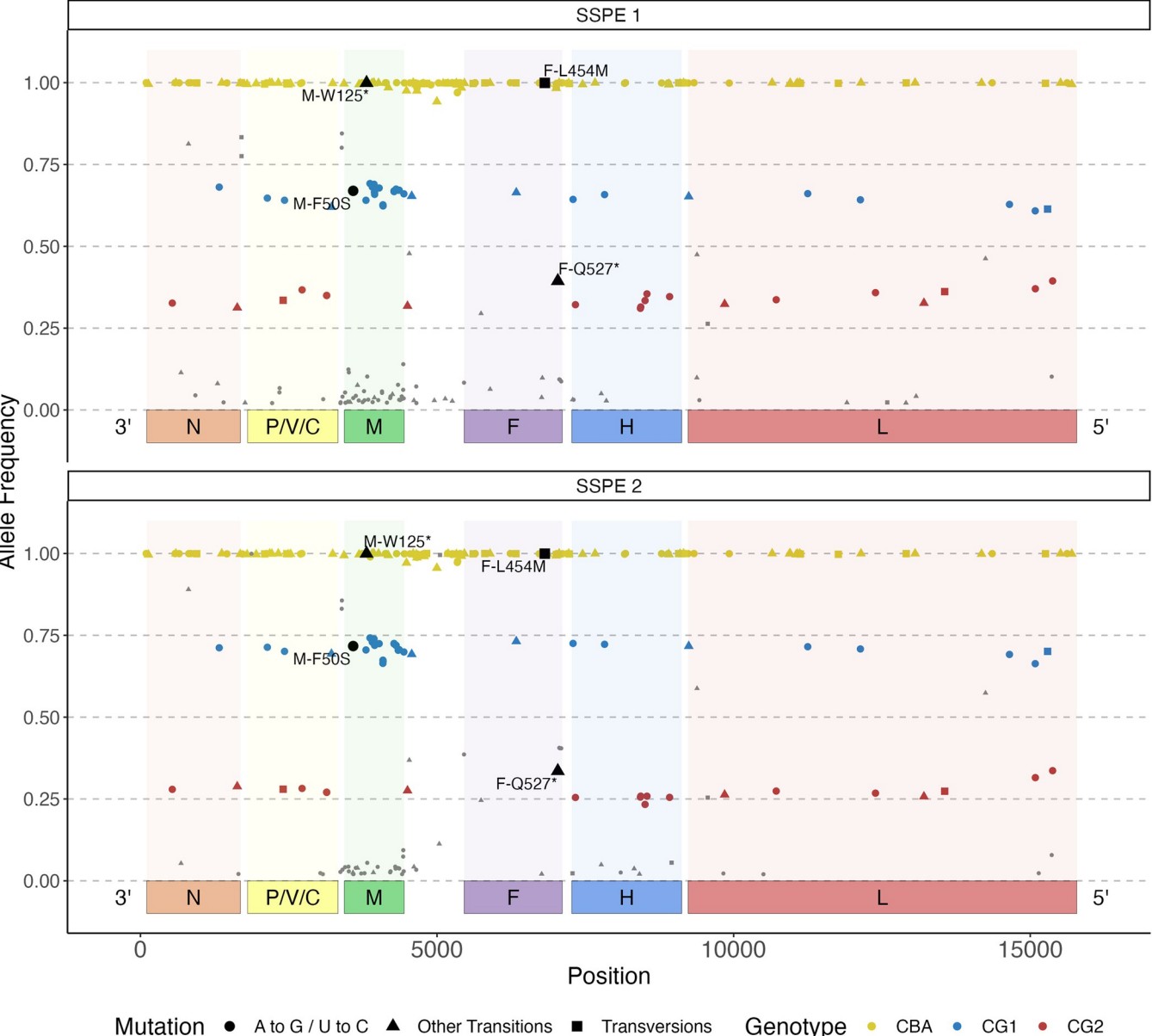

**Fig 2. Frequency and genomic location of positions at variance between the reference genome and the SSPE1 (top) and SSPE2 (bottom) sequences.** x-axis: MeV genome location. y-axis: allele frequency. Nucleotide variants detected at nearly 100% frequency are shown in yellow, those detected at 60–75% in blue, those at 25–40% in red and those at other frequencies in grey. Variants shown in black are candidate neuropathogenesis drivers. Dots represent A to G and U to C transitions that may have been introduced by ADAR1 editing [78,79], triangles represent other transitions and squares represent transversions.

differing between the MeV genotype D reference sequence at each position in the MeV population replicating in both pilot samples. In SSPE1, there were 264 variable positions. Among these, three clear groups emerged; 130 SNVs present at >90% frequency (yellow dots), 35 SNVs at 60–75% frequency (blue dots), and 21 SNVs at 30–40% frequency (red dots). In the SSPE2 sample, we observed the same groups of mutations, but their average frequencies were slightly different at 70% and 30%, respectively.

This data led us to two significant observations about the MeV population in the brain. First, the presence of 130 SNVs that were nearly fixed in the virus population of both specimens suggested that these mutations are ancestral to all the sampled virus sequences. This group of SNVs has been termed the Candidate Brain Ancestor (CBA, including the 130 positions detected at >90% frequency). However, it is important to note that although the presence of these SNVs in both tissues is consistent with the hypothesis that they are ancestral to the virus in the brain, we cannot definitively determine if these variants were acquired before or after brain entry. Second, the presence of two populations of SNVs at congruent frequency in SSPE1 and SSPE2 suggests the existence of two distinct genomes in these specimens. We hypothesize that both specimens had these two distinct viral subpopulations coexisting and that both subpopulations possessed all 130 CBA variants, along with their specific mutations. The subpopulation with the 35 higher frequency variants has been named Candidate Genome 1 (CG1), while the one with the 21 lower frequency variants is referred to as Candidate Genome 2 (CG2).

## Potential drivers of neurotropism acquisition

To focus further analyses, we assessed whether mutations present in proposed sequences CBA, CG1, or CG2 were similar or identical to mutations previously shown to drive brain spread in other SSPE cases. We identified two mutations, M-W125* and F-L454M, fixed on nearly all MeV genomes (black symbols in **Fig 2,** yellow line on top). M-W125* introduces a stop codon interrupting the M protein reading frame after 124 of its 335 amino acids, and F-L454M changes an amino acid that controls the activation energy of the F trimer for cell-cell fusion [33]. Since these two mutations are present at >99% frequency, they were considered to be part of CBA.

Two other potential driver mutations were detected at lower frequencies. F-Q527*, which introduces a stop codon in the F protein cytoplasmic tail, was detected in three other SSPE cases [36]. Since it was at 35–40% frequency, it was assigned to CG2 (**Fig 2,** CG2, black triangle). M-F50S, originally identified in a wild-type MeV variant not linked to SSPE, changes an amino acid that modulates the interaction of M with filamentous actin (F-actin) [45]. Since it was at 65–75% frequency, it was assigned to CG1 (**Fig 2,** CG1, black dot).

## Most cells are infected by both genome populations

Having hypothesized that two distinct genome populations exist, we sought to document how often both genomes are present in the same cell. To accomplish this, we used allele-specific amplified fluorescence *in situ* hybridization (ampFISH), a technique that can discriminate RNA molecules with single nucleotide differences [46]. **S1 Fig** shows a schematic of this method. In addition to using genome-specific probes (ampCG1 and ampCG2), we generated a set of control single molecule fluorescent *in situ* hybridization (smFISH) probes recognizing sequences identical in both genomes (MeV), and stained nuclei by DAPI. The confocal images from 5μm tissue slices from the temporal and occipital lobe (**S2A and S2B Fig**) show that in both tissues both genomes frequently replicate in the same cell. **S2C Fig** shows a negative

**Table 1. Summary of CG1 and CG2 signal quantification in individual cells of three brain specimens.**

| Tissue | Total cells | Candidate Genome 1 percentages | | | | |
|---|---|---|---|---|---|---|
| | | <5% | 5–35% | 35–65% | 65–95% | >95% |
| Temporal lobe | 107 | 5* | 37 | 41 | 24 | 0 |
| Occipital lobe | 100 | 6 | 44 | 34 | 25 | 1 |
| Brainstem | 101 | 0 | 14 | 28 | 52 | 7 |

* Cells with <5% CG1 signal have >95% CG2 signal. Primary data in S3 Fig.

control. Furthermore, successful hybridization provided further evidence for the existence of CG1 and CG2.

We then measured the CG1 and CG2 signal intensities in about 100 cells from three specimens: temporal lobe, occipital lobe, and brainstem. **S3A Fig** shows a confocal image from the temporal lobe, **S3B Fig** shows quantitative data from the marked cells in panel A, and **Table 1** summarizes all the data. In all specimens, co-replication was detected in about 90% of the cells, but the CG1 signal was stronger in the brainstem while the CG2 signal was stronger in the occipital and temporal lobes. Higher resolution analyses identified perinuclear clusters of both CG1 and CG2 replication centers (**Fig 3**, left panel). These genome-specific clusters were occasionally spatially segregated (**Fig 3**, right panel). Thus, CG1 and CG2 co-replicated in about 90% of the cells from three distal brain areas, which indicates frequent co-existence and suggests the possibility of genetic complementation.

## Robust MeV transcription in most forebrain specimens

The high levels of MeV RNA observed in the SSPE1 and SSPE2 specimens were unexpected, as previous studies documented restricted MeV transcription and protein expression in autopsy specimens from other SSPE cases [47,48]. To further characterize MeV transcription in this brain, we thawed it and extracted RNA from 13 specific, spatially distributed regions. Due to thawing, RNA integrity was reduced (**S4 Fig**, top and middle panel). Nevertheless, deep sequencing analysis revealed N to L gradients of transcript abundance in most sampled

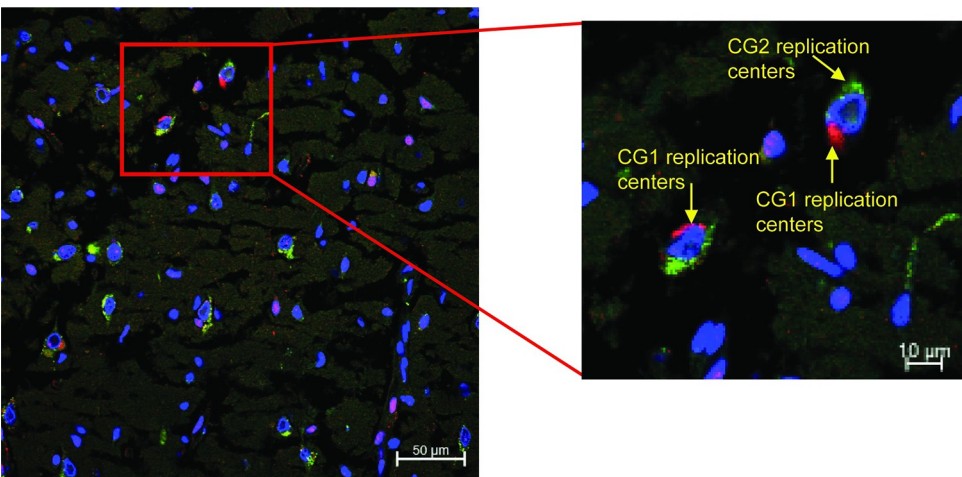

**Fig 3. CG1 and CG2 replicate in the same cells and occasionally form spatially segregated replication centers.** *In situ* hybridization with CG1 (red) and CG2 (green) specific probes in temporal lobe tissue. Nuclei are counterstained with DAPI (blue). Red box highlights the area shown on the right.

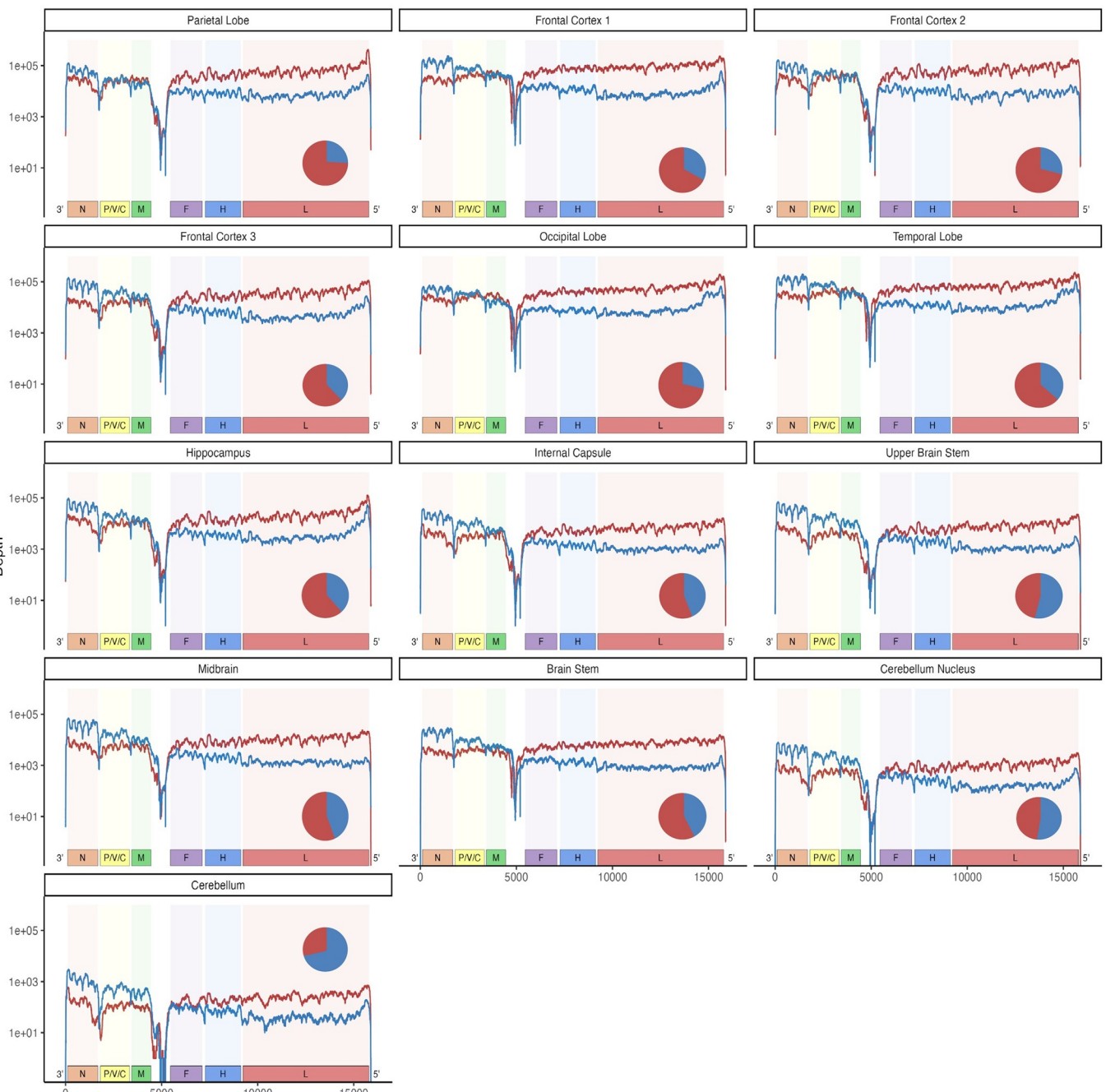

**Fig 4. Distribution of MeV plus and minus reads in brain specimens.** MeV genome coverage plot showing positive (blue line) and negative (red line) strand reads. x-axis: MeV genome; y-axis reads per nucleotide on a logarithmic scale. Pie charts show the ratio of positive (blue) and negative (red) strand reads.

regions, confirming active transcription (**Fig 4**, plus strand reads frequencies shown by blue lines; the vertical axis uses a logarithmic scale).

Total viral RNA levels were high. In two of the frontal cortex specimens and the parietal lobe specimen, the viral reads accounted for 19–20% of the total reads. In six other specimens, between 4–12% of reads were viral. In the internal capsule and brain stem, about 2–3% of reads were viral, and the two cerebellum specimens had fewer than 1% reads mapping to MeV

(**S5 Fig**). Even considering a bias for preferential protection of encapsidated genomes from RNAse degradation during thawing, these data imply robust viral replication in most forebrain specimens.

## Abundant defective genomes in some specimens

Analyses of MeV negative strand-reads indicated that, in some specimens, they were more abundant than the plus strand-reads (**Fig 4**, red lines and circular insets). These analyses also revealed overrepresentation of reads aligning proximally to the genome 5' end in the parietal lobe and hippocampus (**Fig 4**, left column, first and third panel from top). These are the two specimens in which very high levels of 1–2 kb defective genomes were detected by Northern blot analyses (**S4 Fig**, bottom panel). This confirms that short defective genomes abounded in certain specimens.

## Both MeV genome populations are ubiquitous

To determine the distribution of the candidate genomes, CG1 and CG2, across the brain, we expanded our analysis to include data from 13 additional tissue samples. In total, we obtained around 95 million, 2x150bp long MeV reads for an average coverage of 0.89 million reads/base of the 15,894 bases MeV genome (**S1 Table**). The MeV genome reads from these thawed tissues included about 45 times more reads than the SSPE1 and SSPE2 samples (90 versus 2 million, **S1 Table**).

By jointly analyzing variants across all 15 specimens, we were able to discern which mutations in CBA, CG1, and CG2 were specific to SSPE1 and SSPE2 and which mutations were truly ubiquitous in the brain. We used an unbiased approach to cluster all mutations whose frequencies were strongly correlated in each tissue. Mutations on the same viral molecules should appear at roughly the same frequencies across specimens. This method confirmed the existence of CBA, CG1, and CG2 and revealed mutations unique to SSPE1 and SSPE2, suggesting localized differentiation. Nevertheless, the three sequences differed only minimally from those of the candidate genomes. We named these sequences BA (Brain Ancestor), G1, and G2 to differentiate them from those derived solely from SSPE1 and SSPE2 data.

In addition to our frequency-based haplotyping approach, we also sought evidence for G1 and G2 within single sequencing reads. First, we adopted an approach from the haplotyping algorithm CliqueSNV to assess linkage among G1 and G2 SNVs on Illumina reads as either "linked" or "forbidden" (i.e., unlinked) based on the number of reads possessing both SNVs [49]. Although the read length only permitted us to assess the linkage between nearby SNVs, we found strong support for the G1 and G2 haplotypes. Bridging reads nearly always classified pairs of G1 SNVs as statistically "linked", and never classified them as statistically "forbidden." Similar results were found for G2. In contrast, pairs in which one SNV was G1 and the other was G2 never showed statistical linkage, and were also found to be "forbidden" approximately 36% of the time (**S6 Fig**). Second, we obtained longer reads by nanopore sequencing two specimens with adequate RNA preservation: frontal cortex 1 and hippocampus. Analyses of hundreds of sequencing reads spanning 900 or more bases over the M gene brought additional physical evidence of the linkage between the mutations attributed to G1 and G2 (**S7 Fig**).

Using BA as our reference, we identified 535 distinct variants with a frequency above 2% in all samples. **S8 Fig** illustrates the position of each variant on the MeV genome by specimen, and **Fig 5** displays all variants by specimen and frequency. In this figure, G1-related variants are blue, G2-related ones are red, and unlinked variants are gray. This shows that both genomes were found in all brain regions, but their distribution varied considerably. Notably,

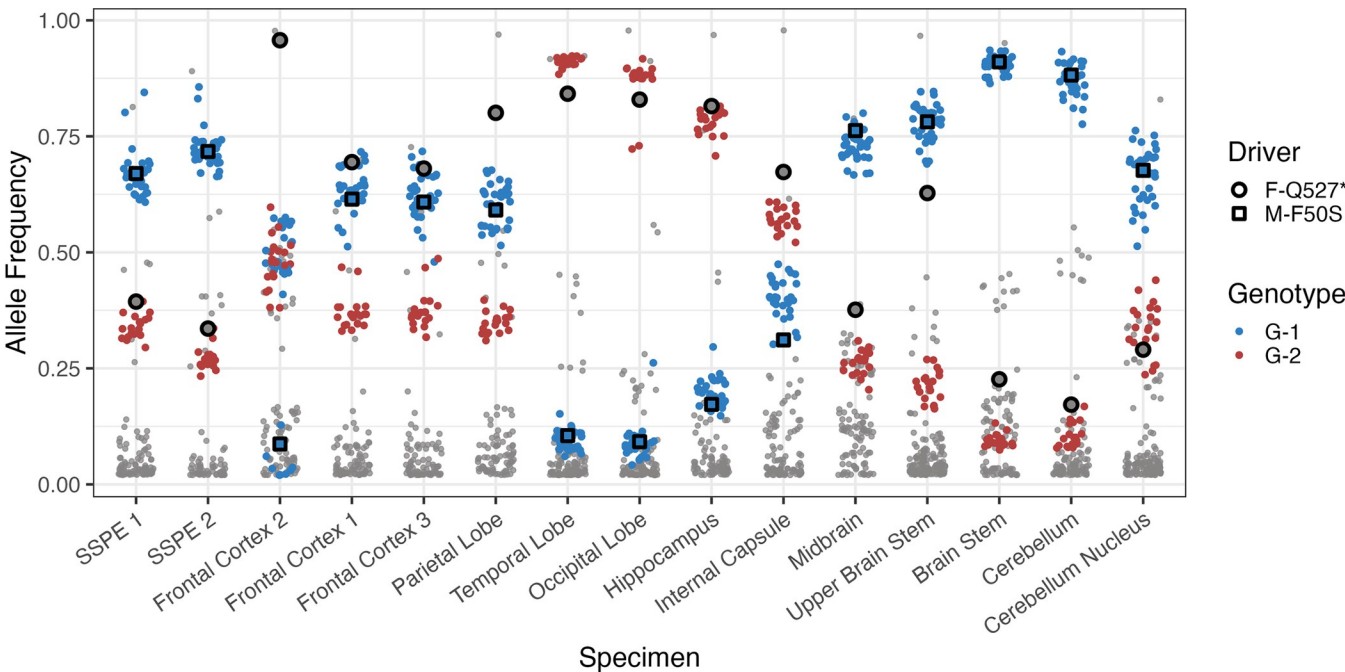

**Fig 5. Frequency of G1 and G2 mutations and two potential neuropathogenesis driver mutations in all brain specimens.** X-axis: brain specimens; y-axis; frequencies of G1 mutations (blue), G2 mutations (red) and all other mutations (grey). Black circles highlight F-Q527* mutations and black squares highlight M-F50S mutations.

except for the case of frontal cortex 2, G1 and G2 mutations appear at comparable frequencies, consistent with the genetic linkage we hypothesized.

## An early G1 ancestor left descendant genomes only in frontal cortex 2

To investigate whether the genetic similarity of the MeV population was correlated with their spatial proximity, we conducted a principal component analysis (PCA) on the frequency of each SNV in each tissue. Rather than focusing on individual sequenced genomes, this approach examines a matrix where each row signifies a spatial location, and every column indicates the SNV frequency in that location. **S9A Fig** shows the relative similarity of each specimen's MeV population given by the first and second principal component, and **S9B Fig** juxtaposes this similarity with the brain location of the specimen from which the RNA was isolated. For the most part, specimens isolated from nearby regions were more likely to have similar MeV populations. One major exception is the frontal cortex 2 specimen, which has a very distinct MeV genome population from neighboring specimens (**S9A Fig**, top).

Closer examination of the frontal cortex 2 specimen revealed two clusters of mutations that separated it from the other samples. One cluster included 10 G1 mutations at substantially reduced frequencies compared to the other G1 mutations (**Fig 5**, frontal cortex 2, bottom), while the other cluster contained 11 mutations that were largely absent from other specimens, yet they were present at nearly the frequency of the remaining G1 mutations in frontal cortex 2. A phylogenetically parsimonious explanation of these observations is that an ancestor to G1, which we call G-01, underwent two divergent evolutionary histories. In one, it acquired a set of 10 mutations, which we call G-01b, to form G1. In another, it acquired a different set of 11 mutations, which we call G-01a, to form a separate genetic background which we call G-FC2 to indicate that it is found nearly exclusively in frontal cortex 2.

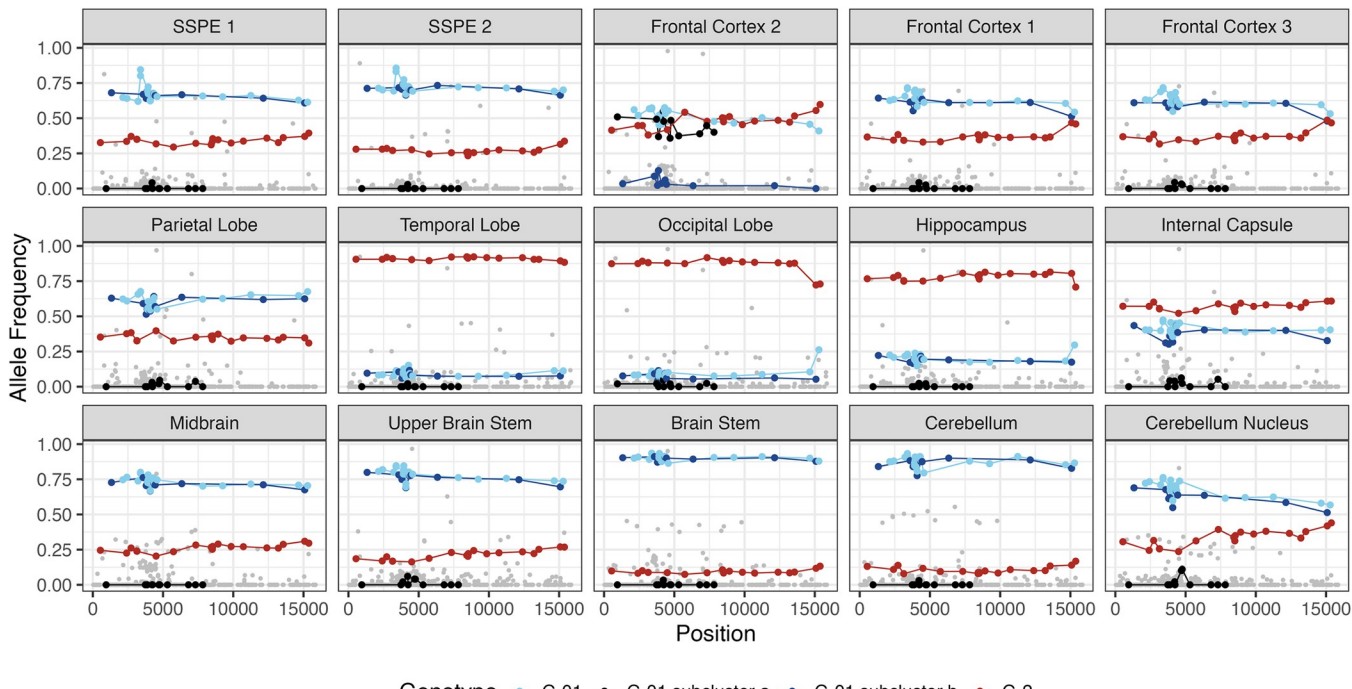

**Fig 6. Identification of a spatially restricted G1 subpopulation in frontal cortex 2.** For each panel x-axis: MeV genome location; y-axis: allele frequency. SNVs attributed to G-01 are shown in light blue and linked with a line. SNVs attributed to G-01b are shown in dark blue and linked with a line. SNVs attributed to G-01a are shown in black and linked with a line. SNVs attributed to G2 are shown in dark red and linked with a line. All other SNVs are shown in grey. SNVs are defined relative to BA.

These observations are visualized in **Fig 6**: G-01b is shown as a dark blue line joining 10 dark blue mutations (where mutations are represented as dots), G-01a is shown as a black line joining 11 black mutations; G-01 is shown as a light blue line joining light blue mutations; and the G2 mutations are shown as red mutations joined by a red line. Except for frontal cortex 2, all G1 (G-01 plus G-01b) mutations are detected at the same frequency in every specimen, suggesting their concurrent spread through the brain on a single genetic background. Among the mutations present in the spatially-ubiquitous G1 genomes but absent in the spatially-restricted G-FC2 genomes is M-F50S, which was previously noted as a potential driver mutation.

## F cytoplasmic tail truncation mutations occur repeatedly and vary in frequency spatially

Frontal cortex 2 has a second anomaly: the frequency of F-Q527* is about 95% (**Fig 5**, black border circles). In contrast, F-Q527* frequency in all other specimens is between 15 and 85% (mean = 58%). Since frontal cortex 2 contains both G2 and G-FC2 at a combined frequency of about 95% (**Fig 5**), the frequency of F-Q527* requires its presence on both genetic backgrounds G2 and G-FC2.

However, in other parts of the brain, we have evidence that F-Q527* is present on background G1 as well. In both the parietal lobe and the internal capsule, F-Q527* is at higher frequency than either G1 or G2. Since G1 and G2 together comprise 100% of the population outside of the frontal cortex, F-Q527* must be on both genetic backgrounds to reach this frequency (**S10 Fig**). Furthermore, the frequency of F-Q527* is far above the frequency of G2 in the temporal lobe, occipital lobe, and hippocampus, providing additional evidence that F-Q527* is present on the G1 background.

Close examination of F-Q527*'s allele frequencies across different spatial locations (**Fig 5**, black border circles) reveal that it is not fixed with respect to either the G1 or G2 background (i.e., there are both G1 and G2 genomes that do not possess the F-Q527* mutation). If a mutation is fixed with respect to a particular genetic background, its frequency must be equal to or greater than the frequency of that genetic background in all locations. F-Q527* is at lower frequency than G1 in the midbrain, upper brain stem, brain stem, cerebellum, cerebellum nucleus and both SSPE 1 and 2, and is at lower frequency than G2 in the temporal lobe. As a result, F-Q527* is present but not at 100% frequency on both G1 and G2. There are two potential explanations for these observations that are described at greater length in the discussion: F-Q527* was either gained multiple times on multiple genetic backgrounds, or F-Q527* was gained on the ancestor of G1 and G2 and was reverted (i.e., F-*527Q) multiple times on multiple genetic backgrounds.

Furthermore, a different F cytoplasmic tail truncation mutation than F-Q527*, F-E526*, also spread in the Internal Capsule and Brain Stem. However, F-E526* was on a different genetic background: among 11941 reads overlapping F-Q527* and F-E526*, 5634 contained only F-Q527*, 1361 contained only F-E526* and 1 contained both. Collectively, these results demonstrate that mutations prematurely truncating the cytoplasmic tail of F arose to detectable frequency multiple times but rarely fixed with respect to their genetic backgrounds. Notably, F-Q527* was observed at intermediate frequency in MeV RNA from three other SSPE cases [36].

### Recurrent mutation on the H cytoplasmic tail

To assess if other mutations elsewhere in the MeV genome showed similar dynamics, we re-analyzed the joint dataset from all 15 specimens. In addition to F-Q527*, the frequency of the 8th residue of H did not correlate well with the frequencies of either G1 or G2 (**S10 Fig**). H is the MeV transmembrane glycoprotein that binds the receptors [37], and H-I8T is a residue of its cytoplasmic tail that interacts with M to control activation of the membrane fusion apparatus, similar to the F cytoplasmic tail [50,51].

As with F-Q527*, H-I8T's frequency analyses reveal its linkage to both genetic backgrounds (**S10 Fig**, segmented black line): in most forebrain specimens its frequency correlates with G1, but in three hind brain specimens (brain stem, cerebellum, and cerebellum nucleus) its frequency correlates with G2. Note that the above frequencies do not require H-I8T to be fixed on either background; instead, they could emerge from H-I8T existing at a lower frequency on both backgrounds simultaneously. H-I8T is at very low frequency in the frontal cortex 2 sample and we also note that the frequencies of F-Q527* and H-I8T are anti-correlated across specimens, although not significantly so (Pearson correlation = -0.27, $p$ = 0.36, **S10 Fig**).

### Spatial dynamics of the collective infectious unit

The variation of mutation frequencies across specimens suggested to us that distinct G1 and G2 subpopulations may diversify locally. We reasoned that we could exploit correlation among groups of lower frequency alleles to reveal secondary haplotypes on the background of G1 or G2 and thus chart this local diversification. This approach mirrors clonal deconvolution methods from cancer genomics and is necessary because the allele frequencies alone do not otherwise reveal the relationships among different correlated groups of SNVs (i.e., are they on the same or different genetic backgrounds).

To clonally deconvolve the MeV samples, we developed a four-step approach that (1) calculates correlations in frequency among groups of mutations across all specimens, (2) clusters mutations with similar frequencies across specimens using k-medoids, (3) applies clonal

deconvolution methods to derive all evolutionary trees that can explain the cluster frequencies across all sampled locations with minimal mathematical constraints on the cluster frequencies (Materials & Methods) [52], and (4) filters candidate trees by retaining only those supported by reads spanning positions with mutations at two loci on distinct mutational clusters. This process identified 12 well-supported clusters of mutations present at similar frequencies across specimens (**S11 Fig** and **S2 Table**).

The left panel of **Fig 7A** shows the evolutionary tree of the 12 clusters: six descended directly from G1 and five directly from G2, reflecting few shared SNVs beyond those on the G1 and G2 backgrounds; the only exception was cluster 1a that descended from cluster 1. **S12 Fig** reports the frequencies of the eight G-01 clusters (top panel, the seven G1 descendant clusters and G-FC2) and the five G2 (bottom panel) clusters in all 15 specimens. **Fig 7B** reports these frequencies for the 13 specimens of known location on a brain drawing; the frequency of each cluster is indicated by the width of a corresponding color-coded slice in the pie chart.

These analyses revealed extensive MeV genome heterogeneity across brain specimens. For example, while frontal cortex 1 and 3, the parietal lobe and five specimens in the lower brain region (midbrain, upper brain stem, brain stem, cerebellum, and cerebellum nucleus) all were dominated by G1, the descendant sub-clusters were different. Frontal cortex 1 and 3 were composed largely of un-clustered G1 descendants and cluster 1 and 1a; the parietal lobe was largely composed of cluster 3; and cluster 2 was the largest cluster in most lower brain regions.

While certain clusters were constrained to localized regions (cluster 5 in the parietal lobe, internal capsule and hippocampus and cluster 6 in the brain stem and cerebellum), others were not (for example, cluster 4 in the brain stem and upper brain stem, and in frontal cortex 1 and 2), suggesting possible longer-range viral dispersal across neuronal connections. Similar results were observed among G2 descendants, with a mixture of locally grouped and widely dispersed clusters. Notably, most clusters were found across multiple locations, suggesting ongoing migration between brain regions after initial spread and local diversification.

## Discussion

Our deep sequencing analysis of MeV RNA from multiple regions of an autopsied brain has provided important insights into the processes that drove lethal panencephalitis. Viral replication was robust: MeV reads accounted for 10–20% of the total cellular reads in the forebrain and for 0.1–5% in the hindbrain. This finding was unexpected because in an SSPE case examined by quantitative *in situ* hybridization, MeV RNA reached only 0.1–1% of the peak level of MeV RNA in Vero cell infections, leading to the suggestion of a specific replication block in the final phase of SSPE [47]. In three other SSPE cases, MeV nucleocapsid gene transcription levels measured by quantitative Northern blots averaged 1–3% of the peak transcription levels in HeLa cell infections [53]. In the forebrain specimens we studied, 12–20% of total ribosomal RNA-depleted reads were viral, a level only 2–3 times lower than at the peak of HeLa cell infection [24]. Thus, viral replication proceeded unhindered.

Four lines of evidence suggest that a MeV collective infectious unit with migratory capacity emerged in the frontal cortex of this brain. First, the forebrain has the highest frequencies of MeV RNA, potentially consistent with the longest residence. Other reports have documented high MeV genome levels in the frontal cortex [54,55]. An alternative hypothesis is that low levels of RNA in the hindbrain reflect longer-term residence and associated depletion of host cells. This appears unlikely because the hindbrain produces and regulates respiratory activities [56], and extensive viral replication in this area may interfere with respiratory rhythm generation essential for survival.

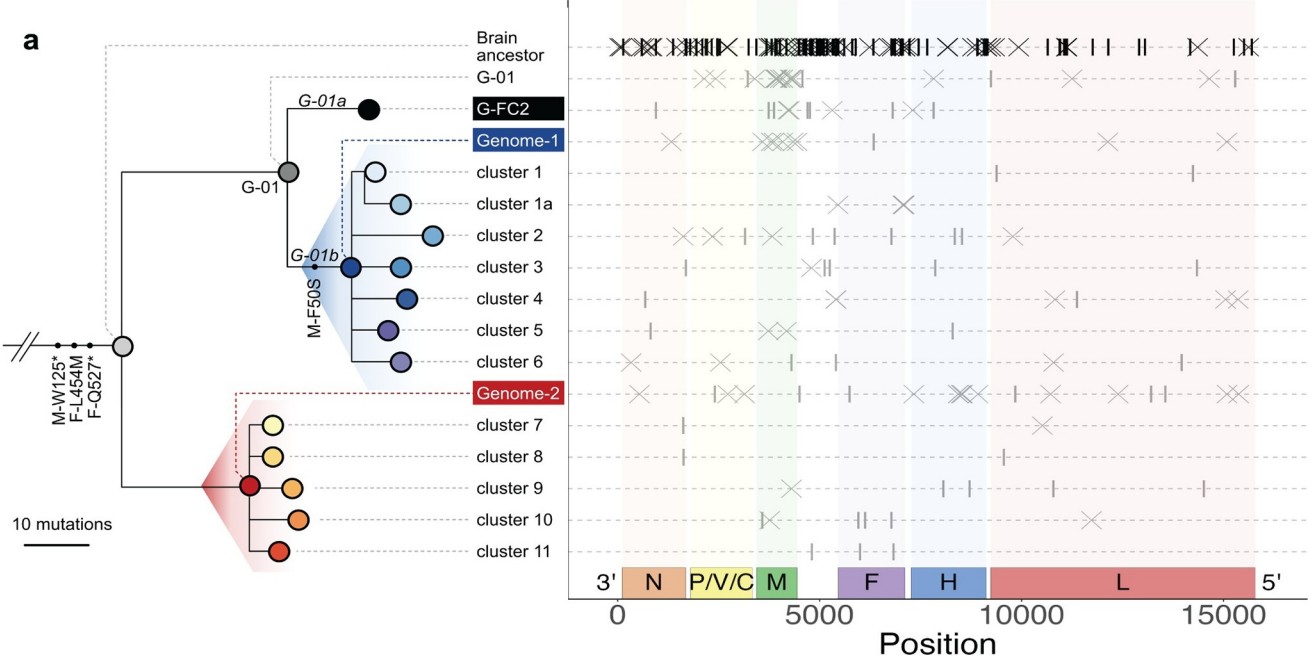

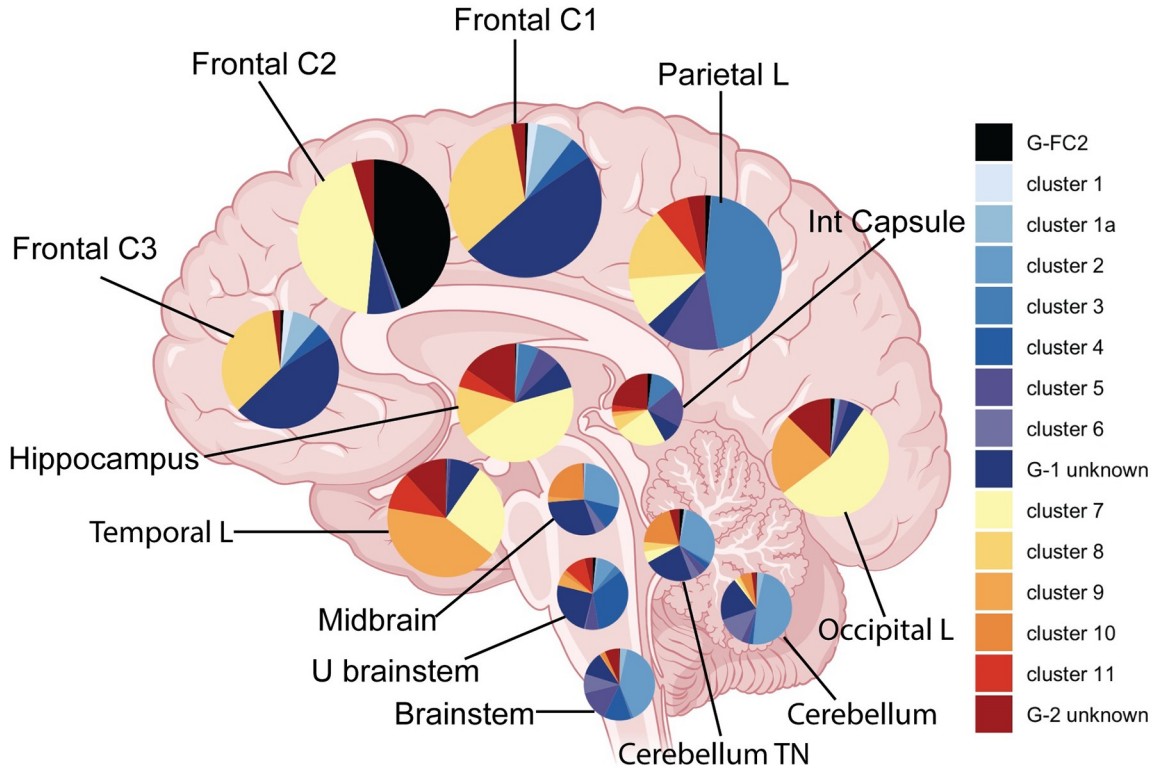

**Fig 7. Spatial dynamics of G1 and G2 subpopulations in the brain.** (**A,** left panel) Phylogenetic tree of G1, G2, and their descendants. (**A,** right panel): location of mutations attributed to the Brain Ancestor, G-01, G-FC2, G1 and its descendants (top), and G2 and its descendants (bottom). Crosses represent A to G and U to C transitions, vertical ticks represent other mutations. (**B**) Brain drawing with superimposed pie charts indicating the frequencies of G1 and G2 descendants. Area of pie chart sectors reflects the frequency of each cluster that are colored according to the key on the right. Large, intermediate, or small pies represent specimens with >13%, 5–13% or less than 5% MeV reads, respectively. C, cortex; L, lobe; U, upper; Int, internal; TN, towards nucleus. Brain image is from BioRender.

Second, we can trace one of the earliest detectable diversification events on the evolutionary tree, that separates G1 and G-FC2, back to the frontal cortex 2 specimen (Fig 7). While it is possible that G-FC2 migrated to the frontal cortex after emergence elsewhere, its strong regional localization suggests it may have limited migratory capacity relative to the G1 and G2 descendant clusters, all of which were found at >5% frequency in two or more specimens. If G-FC2 is non-migratory as its distribution suggests, this links the ancestral population pre-dating G-FC2's emergence to the frontal cortex as well.

Third, descendants of this G-FC2 and G1 ancestor that possess putative driver mutation M-F50S are found at high frequency throughout the brain but at very low frequency in the frontal cortex 2. A potential interpretation is that the genetic background possessing M-F50S emerged at initially low frequency in the frontal cortex and reached high frequency elsewhere due to the founder effect as it colonized new brain regions.

Fourth, the historical branching event leading to the creation of a spatially restricted G-FC2 is challenging to explain if we assume the ancestral MeV initially entering the brain was not capable of brain spread. While we cannot unambiguously determine the site of brain tropism acquisition from the data, the weight of evidence is strongest for a frontal cortex emergence versus any other specific location.

We have constructed a hypothesis of the events favoring MeV genome expansion in this SSPE brain that is consistent with all observed patterns and is illustrated in Fig 8. Because genomes lacking driver mutations were likely constrained to their point of brain entry, the ancestral MeV genome, or collective infectious unit, may have entered the brain in the frontal

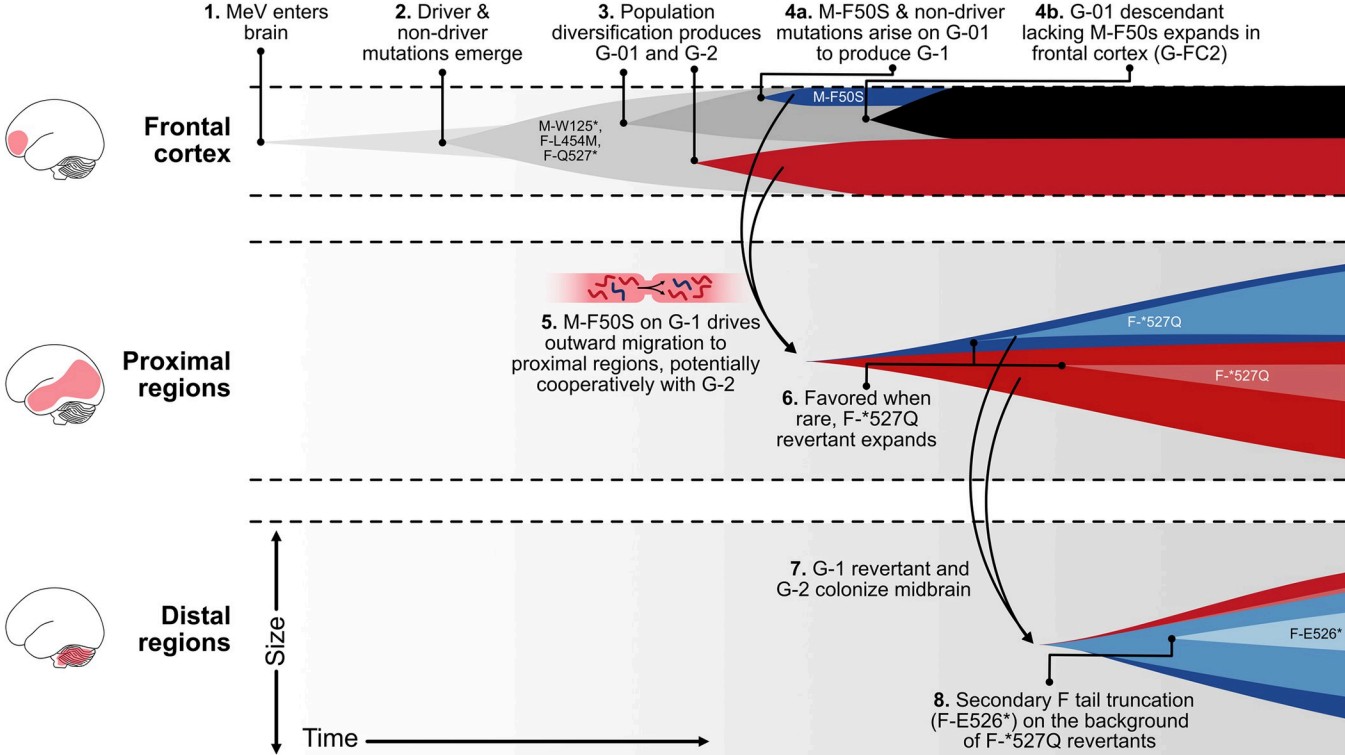

**Fig 8. Hypothetical reconstruction of the evolution of a MeV collective infectious unit in a human brain.** X-axis: time. Y-axis: population size. Cartoon illustrating hypothesized MeV brain expansion over time, including the development of G1 (red), G2 (blue), G-FC2 (black) subpopulations, transit among brain regions, and modulation of F tail truncation. We do not illustrate the simultaneous process of viral diversification forming the G1 and G2 descendant subclusters, or H I8T mutational dynamics.

cortex, possibly via the oropharyngeal route and the olfactory nerve [57] (point 1). Note, that this does not preclude the possibility that MeV entered via one or more additional routes that did not leave descendants contributing to the sampled population.

During replication in the frontal cortex three driver mutations were selected: M-W125*, F-L454M and F-Q527*. This created the genome background on which G-01 and G2 emerged (points 2 and 3; again, we note that other genome variants likely emerged but did not leave descendants that could be detected via sampling). G-01 diversification ultimately produced genome background G1, including the fourth candidate driver mutation M-F50S (point 4a), which spread throughout the brain. Locally in the frontal cortex, a sibling G-01 descendant, G-FC2, came to dominate (point 4b). While these data do not definitively allow us to reconstruct the interactions between G1 and G2 *in vivo*, the addition of M-F50S to the G1 background in the frontal cortex may have enabled both G1 and G2 to migrate outwards as a collective [58,59] (point 5).

The first SSPE clinical signs were detected at age 22 in this patient, suggesting that virus persisted for about 20 years after acute measles. This long incubation period is consistent with very limited spread until the collective infectious unit had accumulated all four driver mutations. We further hypothesize that if either G1 or G2 had begun spreading substantially before the other, we may have been able to locate brain regions infected by one genome but not the other. We did not observe any such brain regions.

As the collective infectious unit began to move outward, the allele frequency of the driver mutation F-Q527* decreased from about 95% in frontal cortex 2 to 60–80% in nearby frontal cortex and lobe regions and as low as 20% in the cerebellum (**Fig 5**). The relative frequencies of F-Q527* in G1 and G2 suggest that a back mutation (i.e., F-*527Q) rescued function on both genome lineages at least once (point 6).

Recurrent mutation is a classical signature of selection, bolstering evidence for F-Q527* as a functionally important position. We hypothesize that the basis of these recurrent back mutations is stabilizing selection towards an intermediate level of fusogenicity. Cooperative interactions between more and less fusogenic MeV variants were recently demonstrated to enhance cell to cell transmission *in vitro* relative to either variant in isolation [60]. Further, allelic heterogeneity at F residue 527 was previously observed in three other SSPE cases from which the F mRNA was directly sequenced, where at least 30% of the sequence was wild type [36]. All these observations are consistent with the hypothesis that revertants in regions proximal to the frontal cortex may have contributed to the spread towards the brainstem (point 7).

In the brainstem the reverted full length F cytoplasmic tail was truncated by a different mutation, F-E526*, further suggesting that an adjusted ratio of full length to truncated F cytoplasmic tails may be critical for spread leading to panencephalitis (point 8). Since the lower brainstem and the hindbrain produce and regulate respiratory activities [56], it is possible that viral replication in these areas interfered with respiratory rhythm generation essential for survival, causing death 14 months after the first clinical signs.

A limitation of our study is that we cannot unambiguously eliminate other explanations of the spatial dynamics of MeV spread in this brain. For example, rather than being gained in an ancestral population and then lost multiple times on multiple backgrounds (G1 and G2), F Q527* could have been gained independently on G-FC2, G1 and G2. Given that numerous mutations can cause premature tail truncation, we consider the independent truncation on multiple genetic backgrounds through the exact same mutation less likely than a single occurrence with independent reversions. Furthermore, F-Q527* is fixed on G-FC2, a variant that was not observed to spread throughout the brain, suggesting that F-Q527* was inherited from an ancestor (G-01) rather than being independently selected on the G-FC2 background. While sampling at autopsy does not allow exact reconstruction of the events driving MeV brain

tropism acquisition, the hypothesis illustrated in **Fig 8** is a parsimonious explanation of all observations.

Other important limitations of our study are that it is confined to the analysis of a single SSPE case, and that it does not include functional analyses of the proposed drivers of neuropathogenesis. However, the MeV genomes replicating in this brain did acquire mutations similar or identical to those previously identified in other SSPE cases. The relevance for neuropathogenesis of two classes of mutations has been confirmed: MeV lacking a functional M protein or with a truncated F protein cytoplasmic tail lost acute pathogenicity but penetrated more deeply into mouse brain parenchyma than standard MeV [41,42]. We have generated recombinant MeV with individual amino acid changes proposed to drive brain tropism acquisition. We are assessing the functional effects of these mutations on the intracellular transport of viral components, and intercellular spread, in neuronal cell lines, compartmentalized primary neural cell cultures, and human brain organoids.

Another limitation of our study is that it is, to our knowledge, the sole analysis presenting data suggesting a key role for collective infectious units in the acquisition of human brain tropism. However, prior sequence analysis of multiple specimens from another SSPE case uncovered evidence of five co-replicating MeV genomes [54]. It was also shown that functional MeV can evolve by co-packaging two genomes, each carrying an F protein unable to mediate membrane fusion on its own, but together exhibiting enhanced fusion activity through hetero-oligomer formation [60]. And it was recently shown that cooperation between genomes coding for wild-type and SSPE-derived mutant F proteins is required for efficient spread in a MeV neuropathogenesis model [61]. Two of the F protein mutations tested in this model, I62T and I446T, were fixed in the BA sequence, and another tested mutation, F-Q527*, varied in frequency.

Taken together, these observations suggest that cooperative interactions of MeV collective infectious units are frequently instrumental for SSPE neuropathogenesis. However, dominant virus variants may continuously evolve in the same brain, and in different brains, a diverse combination of mutations may drive lethal panencephalitis. Accordingly, in this brain we identified mutations previously monitored in other SSPE cases as well as mutations like M-F50S and H-I8T, both expected to impact the respective protein function from previous studies, but not previously identified in other SSPE cases.

It is increasingly appreciated that pathogens can spread as collective infectious units within and between hosts, and MeV is a prime example of this infection paradigm [58,62,63]. After initial MeV amplification in lymphatic organs, virus-infected lymphocytes may deliver multiple genomes to airway epithelial cells [64–66]. MeV genomes spread collectively through localized cell fusion in the airways, as shown in *ex vivo* infections of human airway epithelia [67]. Moreover, expulsion by coughing of infectious centers containing hundreds of MeV genomes may contribute to the extremely high measles reproduction number [68,69]. However, genetic characterization of the spatial dynamics of collective infectious units of MeV and other pathogens in animal hosts is demanding, and it is rarely feasible in humans.

This SSPE case has provided a unique opportunity to gain insights into the spread of a collective infectious unit in a human host: we found that MeV replication can be ubiquitous in the brain and that it can be driven by multiple distinct viral genome lineages that co-colonize even at the single cell level. We present a hypothetical reconstruction of the evolutionary events driving brain adaptation and spread, beginning with probable infection emergence in the frontal cortex and resulting in a genetically diverse and widely dispersed viral population at patient death. We identified putative driver mutations affecting the cytoplasmic tails of both envelope proteins that appear to be independently and recurrently selected across brain regions and genetic backgrounds. These mutations seem constrained to intermediate

prevalence by frequency-dependent selection, which recent experimental results suggest may permit the virus to achieve optimal fusogenicity for brain spread [61]. Re-examination of published data implies that similar selection processes occurred in other SSPE cases. Taken together, these results indicate that collective infectious units can be an important evolutionary unit for MeV brain colonization and raise profound questions about the importance of collective infectious units in human disease.

## Methods

### Patient information

This project was reviewed by the CDC Human Subjects Committee and considered research. It qualified for exemption because the tissue samples were obtained at autopsy from a fatal SSPE case. Disclosure of following patient information was approved. The patient was a 24-year-old US resident who expired in February 2010. The individual, who was born outside of the US, presented with clinical signs consistent with SSPE in December 2008. There was no history of travel, no exposure to measles, and no reports of measles cases in the country of residence, speaking against an acute encephalitis diagnosis.

### SSPE diagnosis and brain specimens

The brain was harvested at autopsy and shipped to the CDC on dry ice. For diagnostic purposes a small section of tissue (approximately 5mm x 5mm) was excised with a sterile scalpel. Following RNA extraction using an RNA Mini-kit (Qiagen), endpoint RT-PCR assays targeting the MeV N gene RNA and Sanger sequencing of the PCR product [69] confirmed the SSPE diagnosis and identified MeV genotype D3. For the pilot experiment, two specimens were collected from the surface of the frozen brain frontal lobe by using a scalpel and tissue punch. When the entire brain was thawed, 13 tissue specimen were collected for RNA extraction and frozen at -70C. Three additional tissue specimens from occipital lobe, temporal lobe and brainstem were fixed with formalin and paraffin-embedded for histological analysis and *in situ* staining. This activity was reviewed by the CDC and was conducted consistent with applicable federal law and CDC policy.

### RNA extraction

Frozen specimens weighed one to two grams. Six to seven ml of Trizol reagent (Invitrogen) was added to each tissue and homogenized using a 150 electric homogenizer (Fisher Scientific). Two ml of chloroform was added to the tissue homogenized in Trizol, vortexed and the resulting approximately 10 ml were aliquoted in 1.5 ml Eppendorf tubes. From here on, Trizol extraction was as per the manufacturer's protocol; RNA pellets from all Eppendorf tubes were pooled and resuspended in diethyl pyrocarbonate-treated water. The RNA concentration was assessed using a Nanodrop 200 Spectrophotometer and samples were stored at -80˚C.

### Northern blots

Three μg of SSPE brain or control RNA were separated on 1% (weight/volume) agarose gels supplemented with 2% (volume/volume) formaldehyde and transferred onto nylon membranes as previously described [44,70]. Northern blot analysis using the DIG-system (Roche) was performed as per the manufacturer's protocol. For detection of N mRNA, a DIG-labelled ssRNA probe N(+) comprising MeV nucleotides 5–254 (GenBank MH144178) was generated by *in vitro* transcription with SP6 RNA polymerase from a plasmid encoding this sequence

under the control of an SP6 promoter. To detect negative strand genomic RNA, a DIG-labelled ssRNA probe L(-) was used as previously reported [44].

### *In situ* hybridization

smFISH probes were prepared from 3'-aminolabeled pooled oligonucleotides as described earlier [71]. ampFISH probes were obtained from integrated DNA technologies (IDT), purified via acrylamide gel electrophoresis and snap cooled as described before [46]. To deparaffinize and hydrate the formaldehyde fixed and paraffin embedded (FFPE) tissue sections, the slides were serially incubated for 10 min at room temperature in Xylene (twice), 100% ethanol, 90% ethanol, 70% ethanol and finally in hybridization wash buffer [46]. After equilibrating the section with hybridization wash buffer, tissue sections were incubated with 30ng of each of the ampFISH probes and 25ng of pooled smFISH probes in hybridization buffer at 37C overnight in a humid chamber. The following day sections were washed thrice with hybridization wash buffer and then incubated with 2ml of 2.5mM of each of the four hybridization chain reaction (HCR) hairpins per 50ml of HCR buffer [46] for 4–5 hours at room temperature. Sections were washed again with hybridization wash buffer and then mounted with either deoxygenated medium [71] or with fluoroshield mounting medium supplemented with DAPI (f6057, Sigma-Aldrich) and imaged using Zeiss LSM 980 and 780. Sequences of the MeV-specific regions of the ampFISH probes are indicated in the **S1 Fig** legend. Complete sequences of all smFISH and ampFISH probes are available upon request.

### Confocal microscopy and quantification of G1 and G2 signal

Confocal microscopy for *in situ* hybridization was carried out using an LSM 980, AxioObserver.Z1/7 microscope. Images were collected using a GaAsP PMT detector with 353, 548, 590 and 650 excitation lasers and a C-apochromat 40x/1.20 W Korr objective. Image processing and analysis were carried out using Zeiss ZEN Lite (Blue edition) version 3.5.

For quantification of signal raw TIFF images for each channel were exported and analyzed in ImageJ. To correct for background noise, signal intensity from uninfected cells was also determined and subtracted from that of infected cells. For each cell, the percentage of G1 and G2 signal was calculated by adding the corrected signal intensity of both channels and dividing individual channel intensity by it.

### RNA library preparation and Illumina sequencing

The concentration and integrity of the RNA were assessed on an Agilent Bioanalyzer DNA 100 chip (Agilent). cDNA library prep was conducted using Illumina TruSeq Stranded Total RNA Sample Prep Kit (Illumina) according to the manufacturer's protocol, which depletes ribosomal RNA. DNA fragmentation of 150 bp and two paired end sequencing (2 x 150) of each library was performed on an Illumina MiSeq or HiSeq4000 platform. The fragment length averaged across all 15 samples was 194 bases with a standard deviation of 87 bases.

### Reference genome

The sequence of the virus that infected the SSPE patient is not known. Since diagnostic sequencing identified a D3 genotype, we generated a reference genome sequence including information from the D3 genomes circulating at the time of infection. Chicago-1 is the best characterized D3 genotype, but for this strain only sequences of five genes are available (GenBank U01977, AF462049, U01980, M81903, M81895). Thus, we supplemented the gaps in the

Chicago-1 genome with available sequences from two other D3 genomes: Illinois for the L gene (AF128246), and Tokyo (GQ376027) for the leader, trailer, and intergenic regions.

## Processing of sequencing reads

Human-aligned BAM files were obtained from the sequencing core for every tissue specimen. These BAM files were converted into unaligned FASTQ files using SamToFastq (version 1.126.0), generating FASTQ files for each specimen (split by read group). Processing of the Illumina sequencing reads from all 15 tissue specimens for variant calling and haplotyping was performed using a Snakemake pipeline that is available on GitHub– https://github.com/jbloomlab/MeV_SSPE_Dynamics [72].

First, the unaligned FASTQ files were trimmed of adaptor sequences using the program fastp (version 0.22.0) [73]. In addition to adaptor trimming, fastp was used to remove reads with an abundance of low-quality bases (> 40% of bases with a phred score < 15). Following quality control, viral reads were extracted from the unaligned FASTQ files by matching 31-base long kmers to the composite MeV reference sequence described above using the program BBduk (version 39.01) (https://jgi.doe.gov/data-and-tools/software-tools/bbtools/bb-tools-user-guide/bbduk-guide/). The percentage of MeV RNA reads that remained after filtering is reported in **S5 Fig**.

After filtering and quality control, the MeV reads were aligned to the composite reference sequence described above using BWA mem (version 0.7.17) [74]. Following alignment, a custom python script was used to make a patient-specific MeV reference genome by incorporating fixed viral mutations into the composite reference genome. Briefly, we used the python/samtools interface pysam (version 0.17.0) (https://github.com/pysam-developers/pysam) to count the number of occurrences of each base for every position in the genome. We only counted bases if they had a phred quality score greater than 25. Additionally, we only considered sites with more than 100 reads covering that position. We considered a mutation fixed with respect to the MeV sequences isolated from the patient's brain if they were present at greater than 90% frequency in 12 or more of the 15 sequenced tissue specimens. These mutations were considered 'ancestral' to the MeV sequences observed in the brain and were incorporated into the patient-specific reference. We realigned the processed FASTQ files to this patient-specific reference genome using BWA mem as we did previously. These aligned BAM files were used in the subsequent variant calling and haplotyping analyses.

For strand base analysis, positive or negative strand reads were filtered from aligned BAM files using samtools view (version >0.1.10) to filter for reads with the alignment flags 163/83 (for positive strand reads) or 99/147 (for negative strand reads). We used samtools depth to calculate the coverage over the MeV genome for positive and negative strand reads.

## Variant calling and filtering

To identify MeV single SNVs with respect to the patient-specific reference described above, we use two variant calling programs–lofreq (version 2.1.3.1) and varscan (version 2.4.0) [75,76]. Where possible, we used the same heuristic filters in each program. The minimum phred score was 25, the minimum read coverage was 200, at least 10 reads needed to contain a given variant, and the minimum SNVs frequency was 2%. If filters could not be applied in either program, we standardized these filters post-hoc in R. We annotated the coding effect of each SNV using the program SnpEff (version 5.1) [77]. Neither insertions nor deletions were included in our analyses.

We called variants from the aligned BAMs as well as BAMs split by the positive or negative sense origin of the reads. There was no appreciable difference between SNVs identified in the

positive or negative sense reads. Therefore, all subsequent analyses were performed on variants identified from the full BAM files.

We then unified the SNVs identified by both lofreq and varscan into a single set of variants for downstream analyses. Roughly 89% of variants were identified by both programs. In consolidating the data from both callers, our intention was to eliminate variants found by only one method to reduce potential false positives. However, we observed cases where a variant was detected by both methods in one tissue, but only by one method in another tissue. Given that these variants were recognized by both callers in certain tissues, they are likely genuine variants. Excluding them based solely on their absence in one method for a specific tissue could lead to false negatives. To address this, we retained all variants identified by both callers in any tissue, even if they were detected by only one method in another tissue. This resulted in a final set of 535 unique nucleotide mutations in the brain.

## Haplotype phasing and processing

To reconstruct viral haplotypes, we used an approach that leverages the fact that we have multiple autopsy specimens isolated from distinct spatial regions in the brain. We expect that mutations present on the same viral molecule–or haplotype–will be present at similar frequencies in each of the sequenced specimens. We took advantage of this correlation in frequency to cluster SNVs that are on the same viral haplotype.

We first took variants that were identified at greater than 2% frequency in all 15 tissue samples. We computed a correlation matrix on the frequencies of these SNVs using the Pearson method. Most variant frequencies were either strongly positively correlated or strongly negatively correlated. We computed a distance matrix from the Pearson coefficients and used k-medoids clustering to partition the SNVs into 3 putative haplotype clusters. The degree of clustering was chosen via visual inspection.

After identifying and clustering SNVs present in every specimen, we extended this analysis to SNVs that were missing from one or more specimens. We partitioned the remaining variants based on their average frequency in each sample. SNVs with higher average allele frequencies are likely to have a larger variance in their frequency across specimens, and therefore true correlations are easier to distinguish from noise. The first bin we used included SNVs present at greater than or equal to 25% allele frequency in at least one specimen. After identifying putative haplotypes using the method described above, we moved on to a second bin comprising variants with frequencies between 5% and 25% in at least one tissue. SNVs that were never identified at greater than 5% frequency in a single specimen could not be clustered with this approach due to the difficulty of distinguishing correlation from noise. The full analysis and a more detailed description of the method can be found in this notebook– https://github.com/jbloomlab/MeV_SSPE_Dynamics/blob/main/results/notebooks/phase-subclonal-mutations.html.

## Assessing physical linkage in Illumina reads

We used a statistical framework adopted from the haplotyping approach CliqueSNV to determine if SNV co-occurrence on individual Illumina reads supported the existence of haplotypes G1 and G2 [49]. Specifically, CliqueSNV asks if two SNVs, $A$ and $B$, are linked by estimating the probability that the number of reads spanning the two loci that contain both $A$ and $B$ is at least the observed number, $O_{AB}$, under the assumption that the $AB$ haplotype is very rare (i.e., frequency below $\tau$). If this probability is low, the $AB$ haplotype cannot be readily explained by sequencing errors, and $A$ and $B$ are classified as linked.

Mathematically, CliqueSNV asks if

$$\Pr(x \geq O_{AB} | T_{AB} \leq \tau) = 1 - \Pr(x < O_{AB} | T_{AB} \leq \tau)$$

$$\leq 1 - \sum_{i=0}^{O_{AB}-1} \binom{n}{i} \tau^i (1-\tau)^{n-i} \leq \frac{0.05}{N}$$

where $T_{AB}$ is the true frequency of the AB haplotype, $n$ is the total number of reads spanning the two loci (regardless of allelic identity) and N is the total number of pairs of sites compared. When this equation is true, SNVs $A$ and $B$ are classified as linked.

Bridging reads can also provide strong evidence that two SNVs occur on different haplotypes. Specifically, CliqueSNV calculates the probability of observing at most $O_{AB}$ reads spanning $A$ and $B$ under the assumption that the $AB$ haplotype is common (i.e., frequency above $\tau$). If this probability is low, the hypothesis of a common $AB$ haplotype is rejected, and $A$ and $B$ are classified as forbidden. Mathematically, CliqueSNV asks if

$$\Pr(x \leq O_{AB} | T_{AB} \geq \tau) \leq \sum_{i=0}^{O_{AB}} \binom{n}{i} \tau^i (1-\tau)^{n-i} \leq \frac{0.05}{N}$$

where all terms are defined as in the previous definition. When this equation is true, SNVs A and B are classified as forbidden. Note, that failure to classify two SNVs as forbidden does not imply that they are linked.

For all putatively G1 or G2 SNVs, we collected all reads across all tissues that bridged any two SNVs and considered all pairs of SNVs with at least 10 bridging reads. Of the 212 SNV pairs evaluated, 138 were composed of two putatively G1 SNVs, 58 were composed of two putatively G2 SNVs and 16 were composed of one putatively G1 SNV and one putatively G2 SNVs. For each pair of SNVs, we tested separately if they were statistically linked and forbidden for $\tau = 0.05$.

## Phylogenetic analysis

After identifying clusters of SNVs forming putative haplotypes, we used the algorithm SPRUCE as implemented in the software tool MACHINA (https://github.com/raphael-group/machina) to find all phylogenetic trees that could explain the genetic relationships between these haplotypes and were also consistent with the average haplotype frequencies across the specimens [52].

In brief, SPRUCE accepts the frequencies of clusters of mutations (partial haplotypes) across multiple samples and exhaustively considers all tree-like relationships between these partial haplotypes. It then systematically eliminates potential trees that violate any of the following three assumptions: (1) if partial haplotype A descends from partial haplotype B, the frequency of A must not exceed the frequency of B in any sample, (2) the total frequency of all haplotypes cannot exceed 1 in any sample, and (3) the genetic relationships among partial haplotypes are the same across all samples. Trees must be constructed in this way (as opposed to classical phylogenetic approaches) because we cannot directly measure full haplotypes–we must infer them jointly with the tree itself. We calculated an inclusive error threshold around each mean haplotype frequency by taking the minimum and maximum frequency of haplotype SNVs in each specimen. There were 36 candidate trees that plausibly described the phylogenetic relationship among haplotype clusters.

To narrow down the space of possible trees, we leveraged reads that bridged segregating loci on pairs of haplotype cluster backgrounds to test whether the co-occurrence of haplotype-

specific SNVs supported linkage between the two clusters [49]. We first applied this approach to assign all haplotype clusters to either the G1 or G2 background. Specifically, for each cluster, we identified all SNVs on the focal cluster with reads overlapping either a G1 or G2 SNV. Because G1 and G2 are mutually exclusive, the absence of a G1 allele implies the presence of a G2 allele. For a given SNV on cluster $c$ in a given specimen $s$, we identified all read counts $x11$, $x10$, $x01$, and $x00$, where $x11$ represents the number of reads overlapping the cluster allele and G1, $x10$ represents the number of reads overlapping the cluster allele and G2, $x01$ represents the number of reads overlapping the non-cluster allele and G1 and $x00$ represents the number of reads overlapping the non-cluster allele and G2. If the cluster allele is on G1, the likelihood of observing the distribution of overlapping reads is multinomially distributed:

$$lik(x_{11}, x_{10}, x_{01}, x_{00}|c \text{ on } G1 \text{ in } s) = \frac{(x11 + x10 + x01 + x00)!}{x11! \ x10! \ x01! \ x00!} f_{G1,11}{}^{x_{11}} f_{G1,10}{}^{x_{10}} f_{G1,01}{}^{x_{01}} f_{G1,00}{}^{x_{00}}$$

where $f_{G1,11} = (f_{c,s} + \varepsilon)/(1 + 4\varepsilon)$, $f_{G1,10} = \max(0, f_{G1,s} - f_c + \varepsilon)/(1 + 4\varepsilon))$, $f_{G1,01} = \varepsilon/(1 + 4\varepsilon)$, and $f_{G1,00} = 1 - f_{G1,11} - f_{G1,10} - f_{G1,01}$, and $f_{G1,s}$ is the frequency of G1 in specimen s and $f_{c,s}$ is the frequency of cluster $c$ in specimen $s$. The frequency of a cluster (or G1 or G2) in a specimen was calculated as the mean frequency of all component SNVs of that cluster. For these analyses, we chose $\varepsilon = 0.01$ to incorporate sampling error in our estimated frequencies. Alternatively, if the cluster allele is on G2, the likelihood of observing the distribution of overlapping reads is given by:

$$lik(x_{11}, x_{10}, x_{01}, x_{00}|c \text{ on } G2 \text{ in } s) = \frac{(x11 + x10 + x01 + x00)!}{x11! \ x10! \ x01! \ x00!} f_{G2,11}{}^{x_{11}} f_{G2,10}{}^{x_{10}} f_{G2,01}{}^{x_{01}} f_{G2,00}{}^{x_{00}}$$

where $f_{G2,11} = \varepsilon/(1 + 4\varepsilon)$, $f_{G2,10} = (f_c + \varepsilon)/(1 + 4\varepsilon)$, $f_{G2,01} = 1 - f_{G2,11} - f_{G2,10} - f_{G2,00}$, and $f_{G2,00} = \max(0, (1 - f_{G2,s} - f_c + \varepsilon)/(1 + 4\varepsilon))$. We then assess the weight of evidence for a SNV on cluster $c$ belonging to a G1 or G2 background across the set of all specimens $S$ based on read overlap via AIC:

$$AIC_{G1} = -2 \sum_{s \ni S} log \ lik(x_{11}, x_{10}, x_{01}, x_{00}|c \text{ on } G1 \text{ in } s)$$

$$AIC_{G2} = -2 \sum_{s \ni S} log \ lik(x_{11}, x_{10}, x_{01}, x_{00}|c \text{ on } G2 \text{ in } s)$$

We can then assign the SNV as supporting assignment of cluster $c$ to G1, G2, or neither via the relative likelihood ratio framework. We tested each SNV on cluster $c$ independently, and assigned a cluster to G1 or G2 if all cluster SNVs supported the assignment. The only cluster unable to be assigned in this way was cluster 12, which had 232 SNV pairs assigned to G1, 5 assigned to G2 and 25 inconclusive. The 30 SNV pairs not supporting G1 assignment were in the highly mutated M region where recurrent mutations are likely, and all 5 SNV pairs supporting G2 were C to T mutations. We therefore assigned cluster 12 to G1.

Using this approach, we were able to filter down the number of plausible trees from 36 trees to only 2 trees. Both trees had identical structures apart from a single prediction that cluster 6 was descended from cluster 2 on one tree but not the other. Using the approach described above with reads that bridged segregating loci on cluster 6 and cluster 2, we were able to show that cluster 6 was not linked to cluster 2 and therefore could not be descended from cluster 2. There were 5422 bridging reads over three pairs of SNVs in cluster 2 and cluster 6 across three tissues (Cerebellum, Cerebellum Nucleus, and Brain Stem) with the highest coverage over both clusters. Of these 5422 reads, 1928 contained cluster 2 SNVs, 435 contained cluster 6

SNVs, and 0 contained both cluster 2 and cluster 6 SNVs. Thus, only a single tree predicted by SPRUCE could plausibly explain the phylogenetic relationship of haplotypes in the brain. The full analysis and a more detailed description of this method can be found in this notebook on GitHub– https://github.com/jbloomlab/MeV_SSPE_Dynamics/blob/main/results/notebooks/filter-spruce-trees.html.

## Supporting information

**S1 Table. Total and MeV reads from deep sequencing.**
(DOCX)

**S2 Table. Frequency of SNVs in different brain regions.**
(DOCX)

**S1 Fig. Schematic of high fidelity ampFISH.** (**Top**) Probes used simultaneously to discriminate CG1 from CG2. The grey regions of the probes bind to the targets. The right and left acceptor probes bind on either side of the region encompassing the SNV. Only one of the donor probes can bind to the SNV region depending on the SNV that is present in the genome. To improve signal strength, we targeted a total of 10 SNVs using four sets of probes for each genome, where all SNVs in the CG1 gave rise to red signals and all SNVs in the CG2 gave rise to green signals. Sequence of grey regions for CG1 probes for SNVs 3907, 3908 and 3912: 5'atatGaacGGcacggaac3' (SNV are capitalized); for SNV 3139 and 3140: 5'gctcaccTTtttccc-gat3'; for SNVs 4087, 4088 and 4090: 5'gtggaccGtGGatgttgc3'; and for SNVs 4309 and 4310: 5'accgattGGggtcttc3'. Sequence of grey regions for CG2 probes for SNVs 3907, 3908 and 3912: 5'atatAaacAAcacggaac3'; for SNVs 3139 and 3140: 5'gctcaccCCtttcccgat3'; for SNVs 4087, 4088 and 4090: 5'gtggaccAtAAatgttgc3' and for SNVs 4309 and 4310: 5'accgattAAggtctt3'. (**Center and bottom**) The binding of the left donor-mut to the CG2 target sequence initiates a strand displacement reaction in the left acceptor that leads to generation of a green HCR signal using Cy3-labeled HCR hairpins H1 and H2. The binding of the right donor-wt to the CG1 genome target sequence initiates a strand-displacement reaction in the right acceptor that leads to generation of a red HCR signal using Cy5-labeled HCR hairpins H3 and H4.
(TIF)

**S2 Fig. Discrimination of CG1 and CG2 by high fidelity ampFISH.** (**A-C**) Confocal images showing nuclei in blue, MeV M mRNA in grey, CG1 in red and CG2 in green. (**A**) SSPE temporal lobe, (**B**) SSPE occipital lobe and (**C**) healthy human cerebral cortex. Individual channels for the yellow boxed areas are shown in the right panels.
(TIF)

**S3 Fig. Absolute and relative amounts of CG1 and CG2 in individual infected cells of a temporal lobe specimen.** (**A**) Eleven infected cells marked for ImageJ analysis; yellow outlines identify the areas analyzed. CG1 signals are in red, CG2 signals in green. Nuclei are counter-stained with DAPI (blue). (**B**) Table reporting intensity levels of the CG1 (red squares) and CG2 (green squares) signals in the 11 cells marked in panel (A). Each square represents 5 intensity units. For calculating the signal intensity of CG1 and CG2 in each cell, we divided either CG1 or CG2 signal by the total signal from both probes. As an example, for cell number 1, CG1 signal intensity is 15 units and CG2 is 20 units so the percentage of CG1 signal in that cell will be 15/35*100 i.e., 42.8% and the percentage of CG2 will be 57.2%.
(TIF)

**S4 Fig. Quality of RNA extracted from thawed brain specimens.** Specimens analyzed are listed above each lane. C, cortex; TN, towards nucleus, L, lobe. **Top panel**: methylene blue

stained RNA gel. The ribosomal 28S and 18S RNA positions are indicated. **Middle panel**: Northern blot probed with N(+) probe detecting positive strand RNA. The N and N-P mRNAs are indicated. **Bottom panel**: Northern blot probed with L(-) probe detecting genomic RNA. DI RNA: short defective RNAs.
(TIF)

**S5 Fig. Percentage of reads in different SSPE brain specimens that map to MeV.** Large, intermediate, or small circles represent specimens with >13%, 5–13% or less than 5% MeV reads, respectively. Anatomically closer brain regions are indicated with the same color circle outlines. L = lobe, C = cortex, U = Upper, Int = Internal and TN = Towards nucleus. Image was generated in BioRender.
(TIF)

**S6 Fig. Assessing linkage of G1 and G2 SNVs in Illumina reads.** Y-axis: proportion of SNV pairs with bridging reads showing a statistically significant effect or not for a given test; x-axis: statistical test determining whether a SNV pair is linked (part of the same haplotype) or forbidden (mutually exclusive). Green indicates statistically significant evidence, whereas gray represents the lack of evidence. The absence of evidence for linkage does not imply that a pair of SNVs is forbidden. The converse is also true, that the absence of evidence for two SNVs being forbidden does not mean that they are linked. 138 G1/G1 pairs were tested, 16 G1/G2 pairs were tested, and 58 G2/G2 pairs were tested.
(TIFF)

**S7 Fig. Read alignment using the longest, highest quality and error corrected [80] reads mapped to the M gene.** RNA from Frontal Cortex 1 (G1 high) and Hippocampus (G2 high) was used for cDNA synthesis using the template switching RT enzyme mix (New England Biolabs) with an N6 TS modified random primer [81]. A single library was generated (samples were barcoded and pooled) using the native barcoding SQK-NBD-114-96 Q20+ sequencing kit (Oxford Nanopore Technologies, ONT). The ONT library was sequenced in a single sequencing run using the high-accuracy base-calling model with a minimum Q score of 10 set on an ONT GridION device using one MinION Flow Cell R10.4.1. Using default parameters for all software, the corrected reads obtained (Frontal Cortex 1: 8,585; Hippocampus: 7,380) were aligned against the M gene using Muscle 3.8.425 in Geneious Prime 2021.1.1. The M-gene mapped reads (Frontal Cortex 1: 477; Hippocampus: 352), were further selected based on coverage of >95% of the M protein coding sequence. The longest reads, namely 57a2eac3-d91e-465f-b823-5cc9c757327f (Frontal Cortex 1) and 25e1cdcf-007f-4260-843b-abd1f4230a30 (Hippocampus) are shown. These reads correspond to the dominant haplotypes in each specimen. Blue SNVs are G1 and red SNVs are G2.
(TIF)

**S8 Fig. MeV mutations at >2% frequency in SSPE brain specimens.** Mutations were called relative to BA. Y-axis: specimen names; x-axis: position of each mutation. Pink blocks show areas where the read depth was too low to confidently call variants.
(TIFF)

**S9 Fig. The MeV genome population from frontal cortex 2 is genetically distinct from those in all other brain specimens.** (**A**) Principal components PC1 (x-axis) and PC2 (y-axis) analysis of MeV genome populations. Five groups of genetically similar specimens are encircled by color-coded lines. (**B**) Brain drawing with superimposed circles of the same color for anatomically close locations. The center of Frontal cortex 2 specimen is indicated in black to mark that its PC analysis position does not reflect its anatomical position. C, cortex; L, lobe; U,

upper; Int, internal; TN, towards nucleus. (B) was generated in BioRender.
(TIF)

**S10 Fig. Dynamic modulation of F and H cytoplasmic tail mutations.** X-axis: brain specimens; y-axis; allele frequencies. The mean frequency of G1 mutations +/- the standard deviation in G1 frequency in each tissue is shown in blue. The same is shown in red for G2 mutations. The solid black line shows the frequencies of F-Q527* in each tissue and the black dashed line shows frequencies of H-I8T.
(TIFF)

**S11 Fig. Correlation by frequency identifies genetically linked clusters of mutations.** X-axis: brain specimens; y-axis; allele frequencies. Each facet is a cluster of SNVs identified by their correlation in frequency across all 15 samples (Materials and Methods). The individual SNVs are represented as black lines. A colored ribbon represents the mean frequency of each cluster +/- the standard deviation in each tissue.
(TIFF)

**S12 Fig. Frequencies of G-01, G2 and their sub-clusters in 15 brain specimens.** x-axis, brain specimens; y-axis, allele frequencies. **Top panel:** frequencies of G-01 (blue line) and its descendants (shaded areas color-coded according to the key on the right). **Bottom panel:** frequencies of G2 (red line) and its descendants (shaded areas color-coded according to the key on the right).
(TIFF)

## Acknowledgments

We thank Jana Ritter, Sherif Zaki, Bettina Bankamp and Raydel Anderson (Center for Disease Control) for preparing the autopsy specimens, and Jesse Bloom (Fred Hutch Basic Science Division) for support in the initial phases of this project. We thank Patricia Devaux and Chanakha Navaratnarajah (Mayo Clinic) for insightful discussions, and Esteban Domingo (Universidad Autonoma de Madrid), Matt Taylor (Montana State University), Bert Rima (Wellcome-Wolfson Institute), Patrick Sinn and Stanley Perlman (University of Iowa) for careful reading of the manuscript draft. This work is dedicated to Martin A. Billeter (1934–2022) and Volker ter Meulen, who pioneered the study of the molecular mechanisms of viral persistence in SSPE.

### Disclaimer

The findings and conclusions in this report are those of the authors and do not necessarily represent the official position of the Centers for Disease Control and Prevention, US Department of Health and Human Services.

## Author Contributions

**Conceptualization:** Iris Yousaf, Ryan C. Donohue, Christian K. Pfaller, Sanjay Tyagi, Paul A. Rota, Alison F. Feder, Roberto Cattaneo.

**Data curation:** Iris Yousaf, William W. Hannon, Sanjay Tyagi, Declan C. Schroeder, Alison F. Feder.

**Formal analysis:** Iris Yousaf, William W. Hannon, Ryan J. Dikdan, Declan C. Schroeder, Wun-Ju Shieh, Alison F. Feder.

**Funding acquisition:** Roberto Cattaneo.

**Investigation:** William W. Hannon, Ryan C. Donohue, Paul A. Rota.

**Methodology:** William W. Hannon, Ryan C. Donohue, Christian K. Pfaller, Alison F. Feder.

**Project administration:** Roberto Cattaneo.

**Resources:** Kalpana Yadav, Paul A. Rota, Roberto Cattaneo.

**Software:** William W. Hannon, Alison F. Feder.

**Supervision:** Sanjay Tyagi, Alison F. Feder, Roberto Cattaneo.

**Visualization:** Iris Yousaf, William W. Hannon, Alison F. Feder.

**Writing – original draft:** Iris Yousaf, Alison F. Feder, Roberto Cattaneo.

**Writing – review & editing:** Iris Yousaf, Ryan C. Donohue, Christian K. Pfaller, Kalpana Yadav, Ryan J. Dikdan, Sanjay Tyagi, Declan C. Schroeder, Wun-Ju Shieh, Paul A. Rota.

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
