## [Decision Letter · Decision Letter 0]

6 Sep 2023

Dear Dr. Cattaneo,

Thank you very much for submitting your manuscript "Spatial dynamics and evolution of a measles virus genome collective that drove lethal human brain disease" for consideration at PLOS Pathogens. As with all papers reviewed by the journal, your manuscript was reviewed by members of the editorial board and by several independent reviewers. In light of the reviews (below this email), we would like to invite the resubmission of a significantly-revised version that takes into account the reviewers' comments.

The general consensus is one of considerable interest in your study and its potential contributions to the field. However, the majority of reviewers recommend that your manuscript requires substantial revisions before it can be considered for publication in its current state.

In particular, both Reviewer #1 and Reviewer #2 have raised substantive technical concerns regarding various aspects of your analysis that must be addressed in the revised manuscript. These concerns are not trivial. As Editor, I share several of these concerns and would urge you to thoroughly address each point raised.

The good news is that the reviewers are eager to see this work published, provided the highlighted issues are adequately resolved. Importantly, it appears that the required changes do not necessitate additional wet-lab experiments but rather call for a more rigorous analysis and presentation of your existing data.

I encourage you to submit a revised version of your manuscript that comprehensively addresses all the reviewers' technical concerns. Please also provide a point-by-point response outlining how you have dealt with each comment. We look forward to receiving your revised manuscript.

Best,

We cannot make any decision about publication until we have seen the revised manuscript and your response to the reviewers' comments. Your revised manuscript is also likely to be sent to reviewers for further evaluation.

Sincerely,

Gustavo Palacios

Guest Editor

PLOS Pathogens

Meike Dittmann

Section Editor

PLOS Pathogens

Kasturi Haldar

Editor-in-Chief

PLOS Pathogens

orcid.org/0000-0001-5065-158X

Michael Malim

Editor-in-Chief

PLOS Pathogens

orcid.org/0000-0002-7699-2064

Dear Prof. Cattaneo,

Thank you for submitting your manuscript to PLOS Pathogens. We have now received evaluations from multiple reviewers. The general consensus is one of considerable interest in your study and its potential contributions to the field. However, the majority of reviewers recommend that your manuscript requires substantial revisions before it can be considered for publication in its current state.

In particular, both Reviewer #1 and Reviewer #2 have raised substantive technical concerns regarding various aspects of your analysis that must be addressed in the revised manuscript. These concerns are not trivial. As Editor, I share several of these concerns and would urge you to thoroughly address each point raised.

The good news is that the reviewers are eager to see this work published, provided the highlighted issues are adequately resolved. Importantly, it appears that the required changes do not necessitate additional wet-lab experiments but rather call for a more rigorous analysis and presentation of your existing data.

I encourage you to submit a revised version of your manuscript that comprehensively addresses all the reviewers' technical concerns. Please also provide a point-by-point response outlining how you have dealt with each comment. We look forward to receiving your revised manuscript.

Best,

Reviewer's Responses to Questions

**Part I - Summary**

Reviewer #1: In “Spatial dynamics and evolution of a measles virus genome collective that drove lethal

human brain disease” Yousaf et al. use high-throughput metagenomic sequencing data to infer the dynamics of a measles virus infection in the brain of one individual. Overall, I found the study to be thoughtfully conducted and well-presented. Although this is just a single case study, and therefore impossible to generalize across SSPE cases, it is a unique dataset that offers several interesting insights into potential trajectories of chronic measles virus infections. However, I would like to see some methodological clarifications, some additional analysis to support the haplotype phasing and some tempering of the way conclusions are being presented.

Reviewer #2: In this study, the authors conducted a comprehensive analysis of MeV RNA from 15 distinct brain regions of an individual who sadly succumbed to SSPE. Utilizing deep sequencing and single-cell level techniques, they unveiled extensive replication occurring in most regions.

The researchers postulated that the initiation of brain spread originates in the frontal cortex. Moreover, they identified two distinct major subpopulations of the MeV genome across all 15 brain specimens. However, the proportions of these subpopulations varied among the different brain samples.

The study also discovered the presence of specific mutations affecting the cytoplasmic tails of the envelope proteins, namely F and hemagglutinin (H). Interestingly, the frequency of these mutations fluctuated across brain regions. The authors argue that the modulation of fusogenicity, crucial for brain spread, is controlled by the proportion of these mutations. This phenomenon may allow the virus to achieve optimal fusogenicity.

While the study holds significant importance, certain aspects require better clarification and/or additional supporting data.

Reviewer #3: In a remarkable experimental tour de force the authors have used deep sequencing to analyze patterns of measles RNA in the entire brain of a patient who died of SSPE 20 years after the initial measles virus infection. The availability of the entire brain allowed characterization of viral genomes present in different parts of the brain and reconstruction of a plausible series of events that occurred after the initial infection, leading to widespread movement of viral genomes throughout the brain.

The findings provide important insights into the processes that drove lethal pan encephalitis and establish an unprecedented framework for the pathogenesis of SSPE that was not possible to construct from previous limited analyses. This description of SSPE will no doubt make its way into textbooks.

The conclusions are generally supported by the data although a certain level of speculation is necessary on the sequence of events, given that what happened in the previous 20 years of incubation had to be inferred. The superb analysis of deep sequencing data from different regions of the brain, combined with single cell RNA analysis provides unique insight into SSPE.

**Part II – Major Issues: Key Experiments Required for Acceptance**

Reviewer #1: 1. Establishing linkage between SNVs present in different parts of the virus genome is a non-trivial task using short read data, especially for viruses, for which there can be many distinct variants co-circulating. Yet, establishing these linkages is critical for many of the analyses presented in this paper and therefore, the authors need to convince readers (and reviewers) that the linkage they describe is accurate.

In general, I found the processes used for haplotype phasing (or at least their descriptions within the “Results” section) to be insufficient. For example, on lines 151-153 CG1 and CG2 seem to have been constructed by simply grouping together all SNVs with similar individual frequency estimates in these two samples, with no validation using within read linkage (as is used to some extent later in the paper). Also, on line 213-214, it is unclear how these composite sequences were generated and on lines 224-225, the reasons different SNVs were assigned to G1 vs. G2 is not clear.

The ”Methods” section includes a more detailed description of the later process for defining clusters, but this needs to be better reflected in the “Results” and the data used to try to deconvolute these clusters needs to be shown; for example, the correlations in the frequencies between variants in different samples and the actual frequencies of variants assigned to each cluster. Also, physical linkage within individual sequencing reads should be used more broadly to test the linkage predictions made by the frequency correlations.

On a related topic, it is unclear to me how and why SPRUCE/MACHINA was used for the generation of the phylogeny in Figure 7a. Line 570: “find all phylogenetic trees consistent with the average haplotype frequencies across the specimens.” What exactly does this mean? What assumptions are being made regarding the relationship between haplotype frequency and phylogeny, and why is this needed, as opposed to simply using the inferred haplotype sequences alone? Has this approach ever been validated for use with viruses?

2. It is unclear whether the SNV frequencies being reported were calculated using both genomic (- sense) and mRNA + antigenomic (+ sense) reads. This should be clarified and the authors should include an analysis to test whether frequency estimates of SNVs are consistent between - sense and + sense reads.

3. There are many areas in which the authors overstate what they can infer from the data being presented. The phrasing needs to be modified throughout to better reflect what can be directly inferred from the data and the uncertainty associated with different interpretations.

a. The authors are clearly interested in the temporal dynamics of chronic measles virus infections, and they attempt to infer these dynamics through phylogenetic analyses. However, the fact remains that they only have samples from a single point in time. Therefore, in general, I found the conclusions regarding the temporal dynamics of the infection to be overstated. Speculation about the temporal dynamics should be saved for the “Dicsussion” and presented with an appropriate level of uncertainty.

b. It is also critical to keep in mind that this paper reports a single case study and therefore the results cannot be generalized to all SSPE cases. Therefore, the authors need to modify language in several places to reflect this. For example:

Line 398: “MeV replication is ubiquitous” to “can be uniquitous”

Lines 398-399:: “is driven by multiple” to “can be driven by multiple”

Line 408: “the genome collective is an important” to “can be an important”

c. Similarly, this paper does not include any functional studies to confirm the roles of specific mutations (or genome collectives) in allowing for spread within the brain. Therefore, conclusions regarding these aspects need to similarly be tempered.

Reviewer #2: The study's findings could benefit from greater clarity and elaboration. The report's strength is notably undermined by the use of non-specific language. Enhancing the report's robustness and clarity can be achieved by incorporating more details on the experimental design.

The study involved the extraction of two tissue samples from unidentified brain regions of a deceased individual aged 24. However, there is a significant lack of information regarding the donor, particularly in terms of relevant background details. It would be advantageous to provide specific information about the timing and location of the individual's infection, if available.

To assess RNA quality, the researchers detected 18s and 28s. It would be beneficial to include the results obtained from running samples on an Agilent Bioanalyzer, if possible. This additional information would offer a more comprehensive understanding of the RNA's integrity.

They then sequenced the RNA after depletion of rRNA and assessed the distribution of reads finding a huge reduction of reads towards the 5’ end. My concern with this is that any conclusion that can be made from the RNA sequencing can only confidently be made for the 3’ end as there are tens of thousands more data points than the 5’ end. I would argue that the number of reads on the 5’ are insufficient. The authors go on to make claims about RNA encoding envelope protein F and hemagglutinin which are in this low coverage region, which to me is not convincing. However, this problem doesn’t seem to be the issue for the samples used later in the study.

In line 103 the authors mention an average coverage of .89 million reads/base but it is unclear if this is for what dataset.

The authors compare the sequenced RNA to a composite sequence consisting of 13 isolates of MeV genotype D, which was the only known genotype circulating in Central America at the time of infection. If the date of infection is known, it would be helpful to include that information. Additionally, reporting an estimation of viral RNA in each sample is crucial to determine whether the sequencing analyses could have been affected by the jackpot effect.

Regarding Lanes 230 to 240, while the use of a composite (PCA) makes sense to some extent, it is necessary to compare the sequenced RNA to each individual sequence genome used in the PCA. Conducting pairwise comparisons would provide insights into the patient's relationship with each genome, highlighting the most and least related ones. Supplementary material could be utilized for presenting this information.

Furthermore, for Figure 2, it is important to include the SSPE2 data alongside the shown SSPE1 data, especially considering its discussion in line 146. It would also be beneficial to provide the raw counts used in Figure two, in addition to the frequency percentages.

When discussing the Candidate Brain Ancestor and Candidates Genotypes 1 and 2, it is crucial to include metadata on the circulating strains at the infection location. For in situ analysis, it would be beneficial to have the breakdown of CG2 cells in different brain regions, preferably presented in a table format like Table 1.

I have reservations about the conclusion that MeV spreads from the frontal cortex to other regions without sufficient evidence. The discussion on viral spread within the brain is lacking. It is possible that MeV starts in the cerebellum and spreads from there. This speculative theory could explain why there are fewer reads from the cerebellum, indicating more cell death and suggesting an earlier infection timepoint. This timeline would be contrary to what is shown in Figure 8. It is important to explore alternative hypotheses or provide clarification to attenuate the conclusion.

Additionally, in Figure 6, they demonstrate the reads from 13 different brain regions, raising the question of why they used SSPE1 reads for comparison with the compound genome.

Lastly, in Extended Figure 4, the numbers do not add up to 100%.

Reviewer #3: (No Response)

**Part III – Minor Issues: Editorial and Data Presentation Modifications**

Reviewer #1: 1. It is unclear to me why the authors are not able to provide an approximate location in the brain from which the pilot samples (SSPE1 and SSPE2) were taken. If the brain was frozen intact, which seems to have been the case, then it should have been clear from which region the samples were collected. This needs to be addressed.

2. I would recommend that the authors reconsider the naming scheme that they use throughout the paper and consider whether all of the abbreviations are needed. It is not very easy for a reader to keep track of the differences between CG1, CG2, G1, G2, G01, G01-a, G01-b, etc.

3. Lines 210-211: “Robust MeV replication facilitated extensive data collection:” I would recommend cutting this statement. It is a somewhat strange, and perhaps misleading, way to introduce the fact that they generated a lot of MeV sequence reads. But an equal number of MeV reads could also be generated from lower titer samples, it would just require more total sequencing depth.

4. Similarly, “because we adopted a more powerful sequencing technology” (Line 217) is not needed. The key is that they generated more sequencing reads, not the technology.

5. Line 232, “similarly” should be “similarity”

6. Line 270: “unlinked to F-Q527*, F-E526*” – please clarify what this conclusion is based on.

7. In general, the “Results” is really a mixture of “Results” and “Discussion”, but lines 272-277 are particularly speculative and would be a better fit for the “Discussion.”

8. Lines 290-291: If you are going to report an insignificant correlation (p=0.36), then you should show the correlation to allow the reader to judge the relevance of the trend.

9. Line 303: Clarify the meaning of “flat phylogeny.” Do you mean that it includes a large polytomy?

10. Please clarify the relevance of “no reports of measles cases in the country of residence since 2009” (Line 427) given that the patient developed symptoms in 2008.

11. Line 430: “a standard assay” – please provide a brief description, not just a citation.

12. Line 449: “comprising MeV nucleotides 5-254” – provide a GenBank (or similar) accession # so that readers can unambiguously find the relevant sequence.

13. Lines 454 and 455: spell out smFISH and ampFISH when first used.

14. Illumina sequencing – please clarify the lengths of the reads being generated. Are they 2 x 150 nt? Please also indicate the average length of the sequenced fragment (which can be inferred from the mapping data). This is important for understanding the limitations of analyses examining the physical linkage of SNVs.

15. Line 486: Why 13 isolates? Is this everything that is available?

16. The methods described on Lines 491-501 appear to have only been used for generating read count tables, and then a different, but similar pipeline was used for the rest of the analyses. Can these read count tables not simply be re-generated using the alignment files from the Snakemake pipeline? This would simplify the methods section and help avoid later confusion.

17. “The raw BAM files from Illumina sequencing” – This statement is confusing. Raw data from an Illumina machine should be in fastq format. A bam file is only generated once the reads have been aligned against a reference. Please clarify what reference was used to generate this BAM and whether there was any filtering based on whether the reads did or did not align to that reference.

18. Please report the breadth and depth of the sequencing data for each sample.

19. Lines 535-536: “we did not benchmark our approach to detect these.” Did you benchmark your approach to detect SNVs?

20. Line 541: “to resolve the SNVs” – how were they resolved?

21. Lines 553-554: Why did you choose 3 as the number of haplotype clusters?

22. Line 555: “Following the success…” – what metric was used to establish success?

23. Line 578: “Because G1 and G2 are mutually exclusive” – what is the evidence to support this? Did you look for G1 and G2 mutations within the same reads to show that this doesn’t occur?

24. Figure 1c – why are results only being shown for SSPE1? It would be better to switch to a stacked bar graph and show the SSPE1 and SSPE2 results side by side.

25. Figure 2 legend – “dots” should be “circles”

26. Figure 3 legend is too brief. Please provide more info about the methods and interpretation of this figure.

27. Figure 4: “upper right” should be “lower right”

28. It would be useful to have a main figure that highlights the locations of the different regions of the brain, similar to ED Fig 4.

29. Figure 6 – why is the term “allele” used instead of SNV?

30. Figure 8 – The figure title needs to make it clear that this is a proposed model for what happened, but that the data presented cannot be used to unambiguously infer everything being shown in this figure.

31. Figure 8 - Please do not include “For details see main text”. Figures (and legends) should be able to stand on their own.

32. ED Figure 1a needs more explanation. It is not intuitive for someone without experience with ampFISH.

33. Associated with ED Figure 1, are there controls that can be shown to demonstrate that this experiment worked as expected?

34. ED Figure 5 – “Frequency is relative to a patient-specific reference” – do you mean that mutations were called relative to the reference? Frequency should not depend on the reference. Also, has this reference sequence been deposited in a public repository?

35. ED Figure 7 – specify the meaning of the width of the red and blue ribbons

Reviewer #2: (No Response)

Reviewer #3: I would like to ask the authors to please define ‘collective infectious units’ and ‘RNA genome collective’ as early as possible in the manuscript.

Would it be possible to determine the effects of amino acid changes observed here in F and M proteins on trans-synaptic spead in neuronal cell cultures? It would be of interest for the authors to mention this possibility in the discussion.

PLOS authors have the option to publish the peer review history of their article (what does this mean?). If published, this will include your full peer review and any attached files.

Reviewer #1: No

Reviewer #2: No

Reviewer #3: **Yes: **Vincent Racaniello
---

## [Decision Letter · Decision Letter 1]

10 Nov 2023

Dear Dr. Cattaneo,

We are pleased to inform you that your manuscript 'Brain tropism acquisition: the spatial dynamics and evolution of a measles virus collective infectious unit that drove lethal subacute sclerosing panencephalitis' has been provisionally accepted for publication in PLOS Pathogens.

Best regards,

Gustavo Palacios

Guest Editor

PLOS Pathogens

Meike Dittmann

Section Editor

PLOS Pathogens

Kasturi Haldar

Editor-in-Chief

PLOS Pathogens

orcid.org/0000-0001-5065-158X

Michael Malim

Editor-in-Chief

PLOS Pathogens

orcid.org/0000-0002-7699-2064

Reviewer Comments (if any, and for reference):

Reviewer's Responses to Questions

**Part I - Summary**

Reviewer #1: I am satisfied with the changes made by the authors and I now support publication. It is clear that the authors carefully considered the feedback they received and I think the revised version is substantially improved.

**Part II – Major Issues: Key Experiments Required for Acceptance**

Reviewer #1: (No Response)

**Part III – Minor Issues: Editorial and Data Presentation Modifications**

Reviewer #1: (No Response)

PLOS authors have the option to publish the peer review history of their article (what does this mean?). If published, this will include your full peer review and any attached files.

Reviewer #1: No

---

## [Editor Report · Acceptance letter]

24 Nov 2023

Dear Dr. Cattaneo,

We are delighted to inform you that your manuscript, "Brain tropism acquisition: the spatial dynamics and evolution of a measles virus collective infectious unit that drove lethal subacute sclerosing panencephalitis," has been formally accepted for publication in PLOS Pathogens.

Best regards,

Kasturi Haldar

Editor-in-Chief

PLOS Pathogens

orcid.org/0000-0001-5065-158X

Michael Malim

Editor-in-Chief

PLOS Pathogens

orcid.org/0000-0002-7699-2064